# Axonal RNA localization is essential for long-term memory

Bruna R. de Queiroz [1], Hiba Laghrissi[1,11], Seetha Rajeev [1,11], Lauren Blot[1,11], Fabienne De Graeve [1], Marine Dehecq[1], Martina Hallegger [2,3,10], Ugur Dag [4], Marion Dunoyer de Segonzac [1], Mirana Ramialison [5,6,7], Chantal Cazevieille [8], Krystyna Keleman [4], Jernej Ule [2,3,9], Arnaud Hubstenberger [1] & Florence Besse [1] ✉

Localization of mRNAs to neuronal terminals, coupled to local translation, has emerged as a prevalent mechanism controlling the synaptic proteome. However, the physiological regulation and function of this process in the context of mature in vivo memory circuits has remained unclear. Here, we combined synaptosome RNA profiling with whole brain high-resolution imaging to uncover mRNAs with different localization patterns in the axons of *Drosophila* Mushroom Body memory neurons, some exhibiting regionalized, input-dependent, recruitment along axons. By integrating transcriptome-wide binding approaches and functional assays, we show that the conserved Imp RNA binding protein controls the transport of mRNAs to Mushroom Body axons and characterize a mutant in which this transport is selectively impaired. Using this unique mutant, we demonstrate that axonal mRNA localization is required for long-term, but not short-term, behavioral memory. This work uncovers circuit-dependent mRNA targeting in vivo and demonstrates the importance of local RNA regulation in memory consolidation.

Localization of neuronal mRNAs to dendritic or axonal terminals has recently emerged as a prevalent mechanism implicated in synaptic proteome maintenance and plasticity[1–3]. Rich repertoires of mRNAs were found in both pre- and post-synaptic compartments of adult mammalian brains[4–6] and shown to undergo selective translation in response to different cues, including LTP/LTD-inducing stimuli[7–12]. While pre- and post-synaptic translation of localized mRNAs was shown to contribute to long-term, protein synthesis-dependent, forms of synaptic plasticity (e.g., long-term potentiation (LTP) or depression (LTD))[13–17], if, where and how synaptic mRNA localization is required for memory establishment in vivo has remained largely unclear. Furthermore, although a few studies have suggested that the populations of localized mRNAs may vary depending on cell types[18], developmental stages[19,20], or learning[21], how the activity of specific neuronal circuits impacts subcellular mRNA localization is unknown.

A well-established mechanism for the subcellular transport of mRNAs involves the recognition of localization elements, most frequently found in 3' UTRs, by RNA binding proteins (RBPs)[22–25]. These RBPs promote the assembly of transport-competent complexes, or transport RNA granules, that recruit molecular motors for long-

[1]Institut de Biologie Valrose, Université Côte d'Azur, CNRS, Inserm, Nice, France. [2]The Francis Crick Institute, London, UK. [3]UK Dementia Research Institute at King's College London, London, UK. [4]Howard Hughes Medical Institute, Janelia Research Campus, Ashburn, VA, USA. [5]Murdoch Children's Research Institute, Department of Paediatrics, Royal Children's Hospital, University of Melbourne, Parkville, VIC, Australia. [6]Australian Regenerative Medicine Institute, Clayton, VIC, Australia. [7]Novo Nordisk Foundation Center for Stem Cell Medicine (reNEW), Murdoch Children's Research Institute, University of Melbourne, Parkville, VIC, Australia. [8]INM, Université de Montpellier, Inserm, Montpellier, France. [9]Department of Basic and Clinical Neuroscience, Institute of Psychiatry, Psychology and Neuroscience, King's College London, London, UK. [10]Present address: Oxford-GSK Institute of Molecular and Computational Medicine (IMCM), Centre for Human Genetics, Nuffield Department of Medicine, University of Oxford, Oxford, UK. [11]These authors contributed equally: Hiba Laghrissi, Seetha Rajeev, Lauren Blot. ✉e-mail: Florence.Besse@univ-cotedazur.fr

distance transport along microtubules[26–29]. The vertebrate ZBP1/Vg1RBP/IGF2BP1 protein, for example, recognizes a so-called "zipcode" sequence located in the 3′UTR of the *β-actin* mRNA, thus promoting its active transport to growing axon tips and dendrites of mature neurons[30–34]. As shown by dynamic live-imaging, bidirectional transport of mRNAs along neuritic branches (or scanning) can be coupled to a more regionalized recruitment of mRNAs to activated synapses (capture) in cultured neurons. Such a local recruitment also depends on 3′UTR sequences and *trans*-acting RBPs[35,36], and was proposed to underlie the tagging of activated synapses essential for memory formation[37,38]. Whether local synaptic mRNA recruitment actually occurs in the context of endogenous memory circuits and in response to LTM-inducing stimuli has, however, remained largely unknown.

Combining transcriptomics, quantitative imaging, and functional approaches, we here explored the regulation and function of axonal mRNA localization in vivo in a set of *Drosophila* brain neurons with central function in learning and memory, the Mushroom Body γ neurons. Through systematic RNA profiling of synaptic fractions, we first identified hundreds of mRNAs that are synaptically enriched. smFISH experiments performed on whole-mount brains uncovered various patterns of mRNA localization in the axonal terminals of Mushroom Body (MB) γ neurons. In particular, they revealed that some mRNAs accumulate selectively in a distal axonal sub-compartment innervated by specific input modulatory neurons, suggesting local accumulation in response to circuit activity. To dissect the mechanistic bases of axonal mRNA localization, we combined transcriptome-wide identification of binding sites with functional approaches, thus uncovering that the conserved RNA binding protein Imp is required for the selective transport of mRNAs to Mushroom Body γ axons. Using a unique mutant in which Imp-dependent axonal mRNA localization is specifically altered (*imp-ΔPLD*), we last probed the requirement of axonal mRNA targeting in courtship memory. *imp-ΔPLD* individuals failed to establish long-term memory while exhibiting normal short-term memory, thus providing in vivo evidence of the importance of local axonal RNA regulation in the consolidation of long-term memories.

## Results

### A diverse repertoire of mRNA species localizes to synapses in the *Drosophila* brain

Although *Drosophila* has long been a key model to study subcellular mRNA targeting, the extent of mRNA localization in the adult brain, as well as the identity of localized mRNAs has remained largely unknown. To identify the population of mRNAs that are targeted to synaptic terminals in *Drosophila* brains, we first optimized a protocol based on differential centrifugation and discontinuous sucrose gradient to isolate synaptosome fractions, starting from adult brain homogenates (Fig. 1A)[39]. This protocol generated a fraction enriched both in soluble presynaptic proteins such as the Cystein String Protein (CSP) chaperone and in membrane-associated proteins such as the synaptic vesicle-associated protein Synaptotagmin-1 (Syt-1) or the T-bar core component Bruchpilot (Brp) (Fig. 1B). This fraction was depleted in nuclear markers such as the Elav RBPs or the Lamin protein. Further validating our purification scheme, analysis of the synaptosomal fraction at the EM level revealed the presence of typical nerve ending structures, characterized by the presence of mitochondria, as well as by an enrichment in synaptic vesicles, some docked to pre-synaptic T-bars (Fig. 1C). RNA sequencing of the recovered fraction revealed that synaptosome RNA content is clearly distinct from that of the initial head lysate (Fig. S1A, B), with hundreds of mRNA species exhibiting a specific enrichment in the synaptosome fraction relative to the initial lysate (879 RNAs enriched with a log2FC ≥ 0.85, *Padj* < 0.05; Figs. 1D, S1C and Supplementary Data 1). Synaptosome-enriched mRNAs code for a variety of proteins, with Gene Ontology (GO) analysis indicating a significant overrepresentation of transcripts coding for mitochondrial

and ribosomal proteins (Figs. 1E and S1D), two categories known to be reproducibly enriched in vertebrate neurites[40]. Transcripts coding for secreted proteins, including components of the extra-cellular matrix (ECM), were also significantly enriched (Supplementary Data 2). Although RNAs encoding synaptic proteins were not overall over-represented in the synaptosomal fractions, a number of mRNAs encoding proteins related to synaptic function and plasticity were found enriched, including regulators of neurotransmission (e.g., *rph/Rph3A*, *csas/CMAS*), signaling molecules (e.g., *lk6/Mknk2*) or regulators of the actin cytoskeleton (e.g., *act42A/Actg1*, *arp3/Arpc3B*). Strikingly, various mRNA species shown to reproducibly localize to neurite terminals in Vertebrates were also found in *Drosophila* synaptosomes (e.g., *arc1/Arc*; *khc/Kif5c*; *robl* and *robl37BC/Dynlrb*; *rps21/Rps21*; *levy/Cox6a*; *act42A/Actg1*)[4,11,40,41]. Together, these results thus indicate that a rich repertoire of mRNA species is selectively targeted to synaptic terminals in the adult *Drosophila* brain. They also uncover evolutionary conservation in the functional classes of localized neuronal mRNAs, as well as in their identity.

### mRNAs localize to the axonal terminals of Mushroom Body neurons

To validate our RNA-sequencing experiment and visualize the distribution of synaptosome-enriched mRNAs in the context of *Drosophila* memory circuits, we then performed smFISH experiments on adult whole-mount brains and imaged a population of neurons known for their function in memory formation: Mushroom Body (MB) γ neurons[42–45]. These neurons project their axons to the ventral side of the *Drosophila* brain, forming a distal bundle termed medial lobe (Fig. S2A).

Candidate mRNAs were chosen such as to sample species encoding different protein classes (components of the translational machineries for *rpl15, rpl23* and *rpl24-like*; extracellular protein for *cg2852*; synaptic proteins for *lk6/Mnk2 and rols/TANC1/2*) and exhibiting different degrees of enrichment (Fig. S1C), as well as based on technical parameters important for the optimal design and use of smFISH probes (e.g., minimal transcript length). When available, GFP protein-trap lines expressing *gfp*-tagged transcripts from the endogenous locus were analyzed using anti-*gfp* labeled probes. Using this latter approach, defined diffraction-limited smFISH spots could be observed both in the cell bodies and in the axons of MB γ neurons for *lk6* (Fig. 2B), *rols* (Fig. 2C), and *csas* (Fig. S2E) transcripts. Such spots were not observed when using *gfp* probes in control *gfp*-negative flies (Fig. 2D), confirming signal specificity. Axonal smFISH spots were also observed for non-*gfp*-tagged transcripts such as *rpl15* (Fig. 2A), *rpl23* (Fig. S2E), *rpl24-like* (Fig. S2B), or *cg2852* (Fig. S2C) when using transcript-specific, indirectly labeled, smiFISH probes (Supplementary Data 3)[46]. Notably, a quantitative comparison of the number of detected axonal RNA spots indicated that the extent of axonal targeting varied significantly from one mRNA species to the other, with *lk6* exhibiting about four to five times more molecules in MB γ axons than *csas*, for example (Fig. S2E). As shown in Fig. 2F, RNA abundance in axons did not correlate with RNA abundance in soma, pointing to the existence of transcript-specific mRNA localization mechanisms. To confirm this hypothesis as well as signal specificity, we then analyzed the distribution of transcripts with relatively high expression levels but no enrichment in synaptosome fractions: *His3.3B*, *Serca*, and *camk2*. While a high number of smFISH spots were observed for these mRNAs in MB cell bodies (Figs. 2F, S2D), no signal was seen in MB γ lobe, further highlighting that axonal mRNA localization does not result from non-specific passive diffusion.

Together, these results thus revealed that selected mRNA species localize to the axon terminals of memory neurons in the adult *Drosophila* brain, raising the question of how such specific localization is achieved.

## Identifying Imp-bound mRNAs that localize to MB axonal terminals

We showed in a previous study that the RNA binding protein Imp, a conserved component of RNA transport machineries[34], is actively and selectively targeted to the axons of MB γ neurons in the adult fly brain[47]. This is manifested by the localization of Imp along the axons of MB γ neurons but not of other neurons (Fig. S3A and ref. 47). To more specifically determine if Imp localizes to synaptic terminals, we here

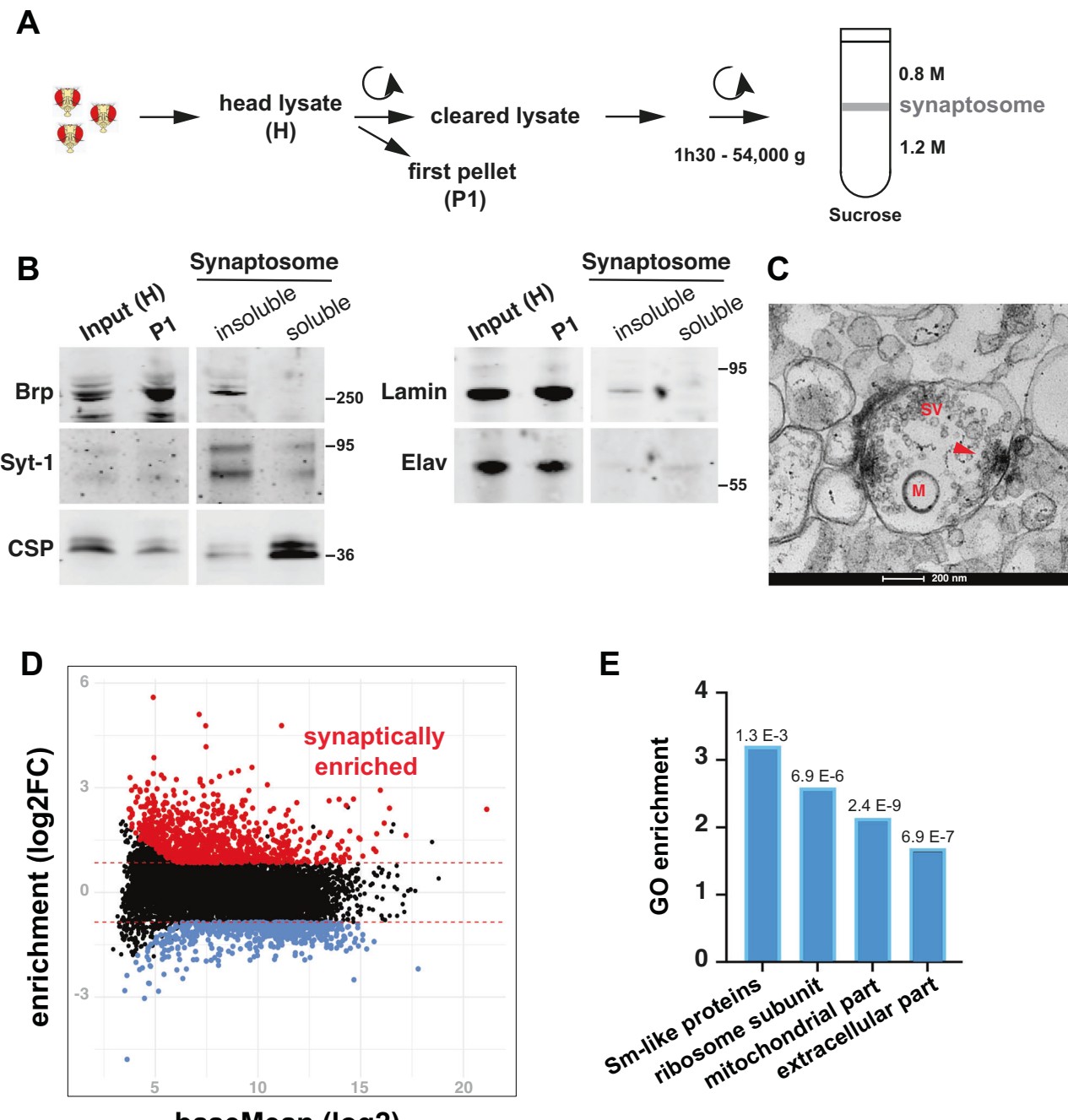

**Fig. 1 | Hundreds of mRNAs are enriched in synaptosome fractions recovered from adult *Drosophila* heads. A** Schematic representation of the protocol used to recover a synaptosome-enriched fraction. See Materials and Methods for more detailed information. **B** Western-Blots performed on the initial head lysates (H, input) and first pellet (P1) (left), as well as on the synaptosome soluble and insoluble fractions (right). Anti-Bruchpilot (Brp) and anti-Cystein String Protein (CSP) antibodies were used to detect membrane-associated and soluble presynaptic proteins, respectively. The synaptic vesicle-associated protein Synaptotagmin-1 (Syt-1) was detected using anti-GFP antibodies on extracts obtained from Syt-1-GFPexpressing flies. Anti-Lamin and anti-Elav antibodies were used to detect nuclear proteins. Blots were performed for each marker from two independent replicates, with similar results. **C** Electron-microscopy image of the synaptosome fraction. Typical presynaptic terminal structures containing synaptic vesicles (SVs), mitochondria (M), and T-bars (arrowhead) are observed. One biological sample was imaged. **D** MA-plot showing the relationship between abundance (reads per million, *x*-axis) and synaptosome enrichment (*y*-axis, log2FC). Transcripts enriched with a log2FC ≥ 0.85 and a *Padj*-value < 0.05 were considered as synaptically-enriched (red data points). Transcripts with a log2FC ≤ −0.85 and a *Padj*-value < 0.05 were considered as synaptically-depleted (blue data points). **E** Main GO component categories enriched in synaptosome fractions (see Supplementary Data 2 for a complete list and associated transcripts). Values above the bars correspond to enrichment test *Padj*-values. Enrichment was calculated in Gorilla using standard one-tailed hypergeometric tests and default parameters. FDR was used for multiple comparisons. Data are provided as a Source Data file.

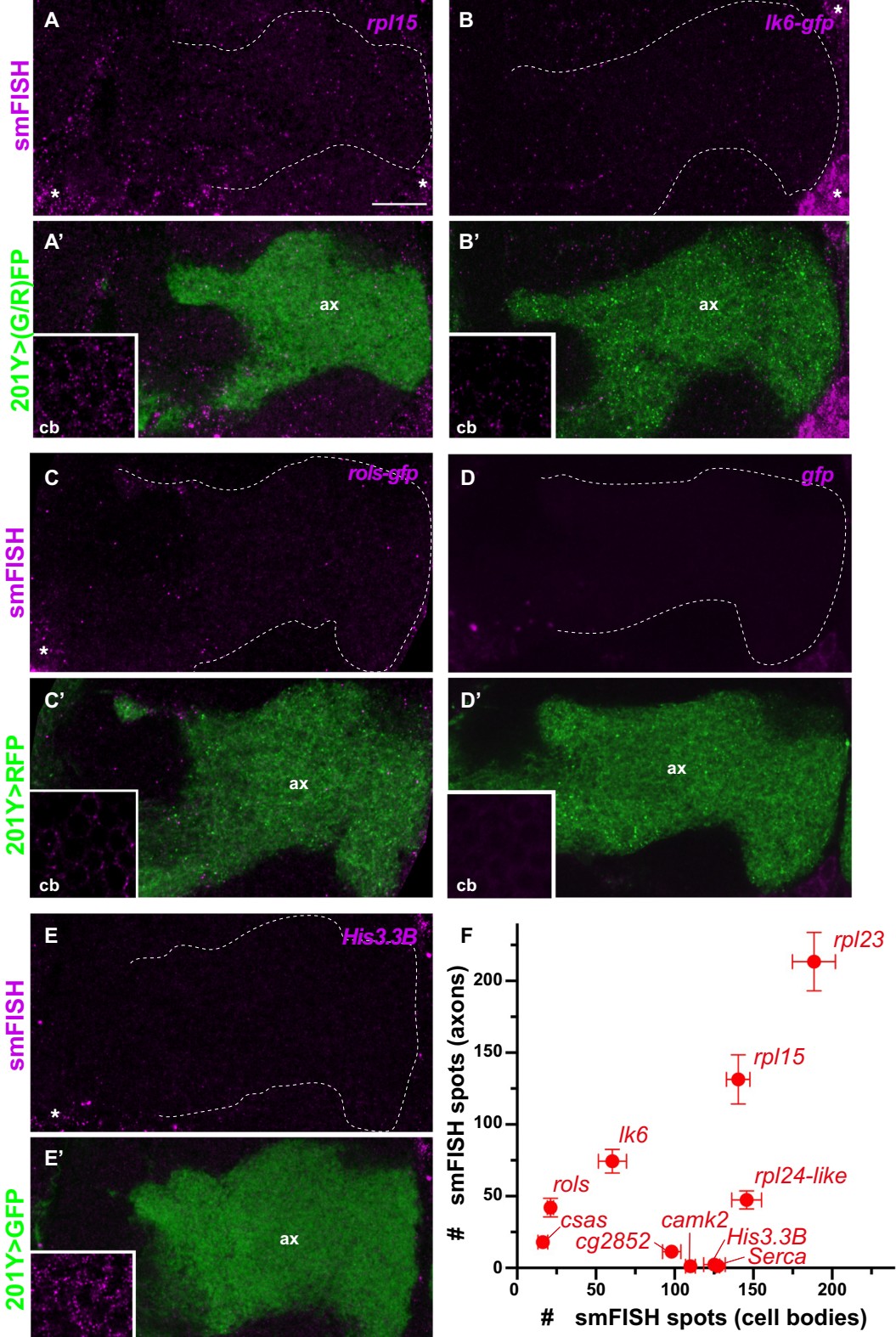

**Fig. 2 | mRNAs localize to Mushroom Body γ axons.** Confocal images of smFISH signals (magenta) obtained from whole-mount adult brains using anti-*rpl15* (**A**), *gfp* (**B**–**D**), and *His3.3B* (**E**) probe sets. The population of MB γ axons (ax) is labeled in green in the bottom overlay images. Boxed areas correspond to single confocal sections of MB γ neuron cell bodies (cb). In (**B**, **C**) GFP protein-trap knock-in lines were used to detect *lk6* and *rols* mRNA distribution. In (**D**) control flies in which GFP is not expressed were used to validate *gfp* probe specificity. MB γ axons were labeled using the 201Y-Gal4 driver in combination with either UAS-cGFP (**A**, **E**) or UAS-CD8-RFP (**B**–**D**). Asterisks indicate signals corresponding to the soma of neighboring neuronal populations. Scale bar: 10 μm. **F** Graph plotting the correlation between the number of smFISH spots in MB γ cell bodies (*x*-axis) *vs* axons (*y*-axis). Data points correspond to mean values and error bars to SEM. Samples collected from two (*rpl23, His3.3B, Serca, camk2, rols, csas*) to three (*cg2852, lk6, rpl15, rpl24-like*) biological replicates, were quantified for each condition. Source data are provided as a Source Data file.

searched for the presence of Imp in adult brain synaptosome fraction (Fig. S3B), which revealed a weak yet reproducible signal consistent with population-specific synaptic recruitment. Together, these findings suggested that Imp may promote the transport of target mRNAs to MB γ presynaptic terminals. To identify these mRNAs, we then aimed at uncovering Imp-bound RNAs enriched in synaptosomal fractions and combined two complementary transcriptome-wide approaches: RNA-immunoprecipitation (RIP) and cross-linking and immunoprecipitation (iCLIP)[48]. RIP was performed from adult heads expressing tagged Imp proteins in MB γ neurons (201Y-Gal4 > UAS-Flag-Imp), where full-length co-precipitated RNAs were isolated, processed for hybridization on GeneChips, and signals compared to those obtained with 201Y-Gal4/+ control samples, thus identifying mRNAs bound to Imp in the population of MB γ neurons. Improved iCLIP was performed from whole adult heads expressing functional GFP-Imp fusions from the endogenous locus (G080-GFP-Imp protein-trap line)[47], thus identifying the precise peaks of Imp crosslinking on RNAs using the iCount peak-calling approach (https://icount.dev/). Two hundred forty-two RNAs were identified as enriched in Imp-RIP experiments (log2FC > 1; Supplementary Data 4) and 1577 in Imp iCLIP experiments (RNAs with at least 1 peak supported by at least 5 uniquely mapped cDNAs; Supplementary Data 5). These included the previously characterized profilin (chickadee) mRNA[47,49] as well as actin5C, the Drosophila ortholog of β-actin mRNA, a well-characterized target of the vertebrate ZBP1/Vg1RBP orthologous protein[31,32,34,50].

By overlapping the mRNAs identified by RIP and iCLIP with those enriched in the synaptosome RNA-seq, we identified candidate mRNAs whose localization to MB γ axons may depend on Imp (Fig. 3A). Among those were profilin, actin5C and arc1, which encode actin cytoskeletal regulators, eIF4A, eIF4E-1 and pabp, which encode conserved translational regulators, as well as other mRNAs encoding proteins with various functions (e.g., collagen α-chain (bnb), transcription factor (crc). The previously validated axonal mRNA lk6 was also identified as an Imp-bound mRNA. smFISH experiments performed on adult brains revealed that these transcripts are all targeted to MB γ axons (Figs. 3B–D and S3C, Supplementary Data 3). Axonal targeting of profilin, eIF4E-1, actin 5C, bnb, and crc mRNAs was weaker than that of arc1, eIF4A, cg15098, and pabp mRNAs (Fig. 3B), yet specific, as revealed by MB-specific, RNAi-mediated, inactivation of profilin (Fig. S3D).

Together, these results thus uncovered a number of mRNAs that are bound by Imp and are localized to the axon terminals of MB γ neurons.

## Imp-bound mRNAs are targeted to MB γ axons through their 3′UTR

The iCLIP methodology enables the precise mapping of RBP binding sites (identified as peaks). To get a transcriptome-wide view of Imp binding preferences, we thus performed a metatranscript analysis, plotting the Imp iCLIP sites along the 5′UTR, coding sequences, and 3′UTR of a normalized reference transcript. This revealed a very strong overall preference for Imp binding to 3′UTRs (Fig. 4A, B), consistent with previous results obtained in other cell types or species[49,51,52]. Analyzing the distribution of Imp iCLIP peaks along the gene regions of profilin and other transcripts confirmed this view, highlighting the near-exclusive binding of Imp to 3′UTR sequences (Figs. 4A and S4). In the case of profilin mRNA, this also pointed to a preferential binding of Imp to proximal and central 3′UTR regions than to most distal regions (Fig. 4A), suggesting the existence of isoform selectivity.

Localization elements, i.e., cis-regulatory regions promoting subcellular mRNA targeting, have for the vast majority of neurite-localized mRNAs been identified in 3′UTR regions[22–25]. To determine the role of 3′UTRs in the localization of Imp-bound mRNA species to MB γ neuron terminals, we thus analyzed the distribution of reporter RNAs in which transcript-specific 3′UTR sequences were cloned downstream of the GFP coding sequence. As shown through smFISH experiments

performed using anti-gfp probes, the 3′UTR sequences of lk6 (isoform RA), pabp (isoform RB), profilin (isoform RB), and actin5C (isoform RC), but not that of a SV40 negative control, were sufficient to target gfp RNA to the axons of MB γ neurons upon expression via the Gal4/UAS system (Figs. 4C, D and S5A).

These results thus suggest that axonally-localized Imp-bound mRNAs are targeted to axonal terminals through specific elements localized in their 3′UTR sequences.

## Compartment-specific recruitment of Imp mRNA targets

Careful analysis of the distribution of Imp mRNA targets in MB γ lobe revealed that a number of these mRNAs (act5C, arc1, pabp, eIF4A, prof, cg15098) tend to exhibit a non-homogenous distribution along axons, characterized by a higher density of transcripts in the most distal part of the MB γ lobe termed γ5 compartment (Fig. 5A–C). Such a behavior was not observed for the abundant, non Imp-bound mRNAs rpl24-like and rpl23. To investigate whether mRNA enrichment in the γ5 compartment depends on 3′UTR sequences, we compared the density of gfp smFISH spots in the γ5 compartments vs more proximal γ2-4 compartments for the gfp-3′UTR transgenes expressed in MB neurons. As shown in Fig. S5B, gfp mRNAs showed distal enrichment, although to a lower extent than endogenous mRNAs, suggesting that other factors than 3′UTR sequences (e.g., 5′UTR sequences, RNA processing …) might additionally contribute to compartment-specific enrichment.

MB γ5 compartment is the most distal of the five anatomical and functional regions (γ1 to γ5) that were defined along the MB γ lobe based on their specific innervation patterns and their capacity to establish and store distinct temporal memory traces[53,54]. It receives input from a specific population of dopaminergic neurons, the PAM-γ5-DANs, and establishes synapses with the glutamatergic MBON-γ5β'2a output neuron (Fig. 5A)[53,55,56]. γ5 mRNA enrichment suggested that selected mRNAs may be recruited to distinct axonal subcompartments in response to the activity of the local circuits they are engaged in. To test this hypothesis further, we compared the distribution of arc1 mRNA in control flies and flies in which the activity of the innervating PAM-γ5-DANs was inhibited through the expression of the inward-rectifying K+ channel Kir2.1[57]. While the overall number of arc1 mRNA smFISH spots found in MB γ axons did not significantly change upon inactivation of PAM-γ5-DANs (Fig. S5C), the observed enrichment in MB γ5 compartment was abolished (Fig. 5D), thus demonstrating that the activation of compartment-specific circuits is required to trigger the local accumulation of mRNAs.

## The axonal localization of selected Imp-bound mRNAs is disrupted in the imp-ΔPLD transport mutant

Having identified Imp-bound mRNAs that localize to MB γ axons actively, and in a 3′UTR-dependent manner, we wondered if Imp was essential for their axonal localization. In a previous study, we showed that the dynamic transport of Imp RNP granules to MB γ axons is impaired in flies lacking the prion-like domain (PLD) of Imp[58]. We also described that in these homozygous viable flies, RNP granule assembly is not altered, and the essential functions of Imp are preserved[58]. To further confirm that the PLD of Imp does not affect RNA binding, we here compared the RNA binding profiles of the wild-type and Imp-ΔPLD proteins in iCLIP experiments performed from GFP-Imp and GFP-Imp-CRISPR-ΔPLD adult heads. This first revealed a very similar extent of iCLIP cDNA counts for Imp and Imp-ΔPLD on 3′UTRs of the vast majority of bound mRNAs (Fig. 6A and Supplementary Data 6), with a few exceptions (Fig. S6A, C, D). Moreover, the profiles of iCLIP crosslink signals of Imp and Imp-ΔPLD were very similar on specific mRNAs, as illustrated for profilin (Fig. 6B) and actin5C (Fig. S6B). To explore if Imp PLD contributes to more subtle fine-tuning of RNA sequence preferences, as previously found for the disordered region of TDP-43[59], we first identified the hexanucleotide (6-mer) motifs that are most

enriched around Imp cross-linked sites and found that most of the top 6-mers contained the YAAY consensus (Fig. S7A). U-rich motifs that generally contained only a single A were also observed (Fig. S7A). We

then compared occurrence of each group of similar 6-mers in wild-type and Imp-ΔPLD data centered on the iCLIP cross-linked sites, which identified similar motif enrichments in wild-type Imp and Imp-ΔPLD

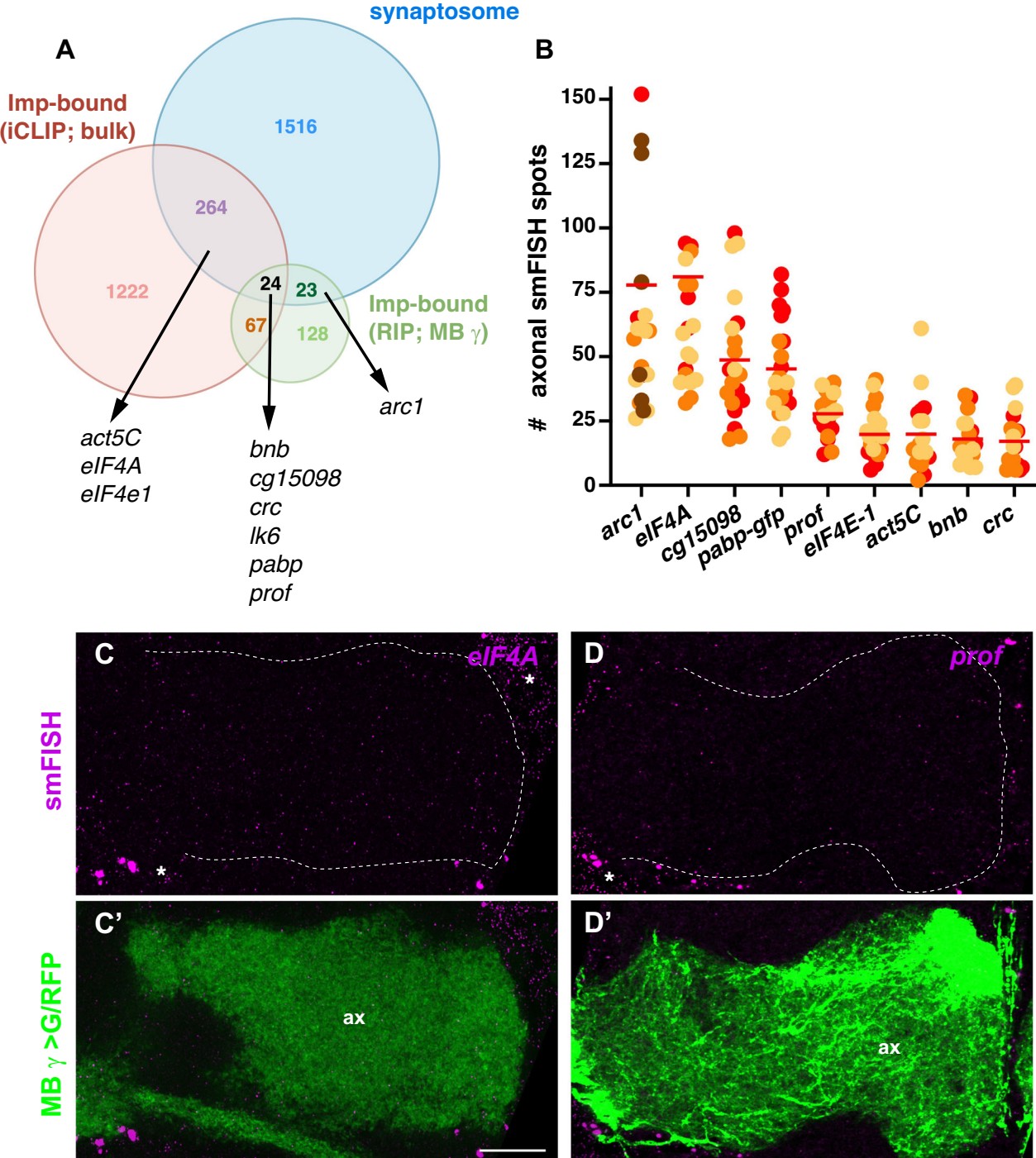

**Fig. 3 | Identification of Imp mRNA targets localizing to MB γ axons.**
**A** Overlapping the data sets from Imp iCLIP (bulk; RNAs with at least a significant peak supported by at least 5 uniquely mapped cDNAs), Imp RIP (MB γ neuron-specific; RNAs enriched with a log2FC > 1) and synaptosome (bulk; RNAs with a log2FC > 0) led to the identification of candidate Imp mRNA targets that may localize to the axons of MB γ neurons. **B** Quantification of the number of smFISH spots in MB γ axons for the candidate Imp-bound mRNAs analyzed. Three to four biological replicates were performed, and data from each replicate was labeled with different colors. Bars represent mean values. *pabp* mRNA distribution was analyzed using a GFP protein-trap knock-in line. Four outlier data points (two for

*arc1* and two for *eiF4A*) were omitted from the graph but considered to calculate the mean. Numbers of brains analyzed: 24 (*arc1*), 21 (*eIF4A*), 21 (*cg15098*), 22 (*pabp-gfp*), 19 (*prof*), 22 (*eIF4E-1*), 18 (*act5C*), 22 (*bnb*), 18 (*CRC*). Confocal images of smFISH signals (magenta) obtained from whole-mount adult brains using anti-*eIF4A* (**C**) and anti-*profilin* (**D**) probe sets. The population of MB γ axons (ax) is labeled in green in the bottom overlay images. MB γ axons were labeled using the 201Y-Gal4 (**C'**) or VT44966-Gal4 (**D'**) drivers in combination with UAS-CD8-RFP (**C'**) or UAS-CD8-GFP (**D'**). Asterisks indicate signals corresponding to the soma of neighboring neuronal populations. Scale bar: 10 μm. Source data are provided as a Source Data file.

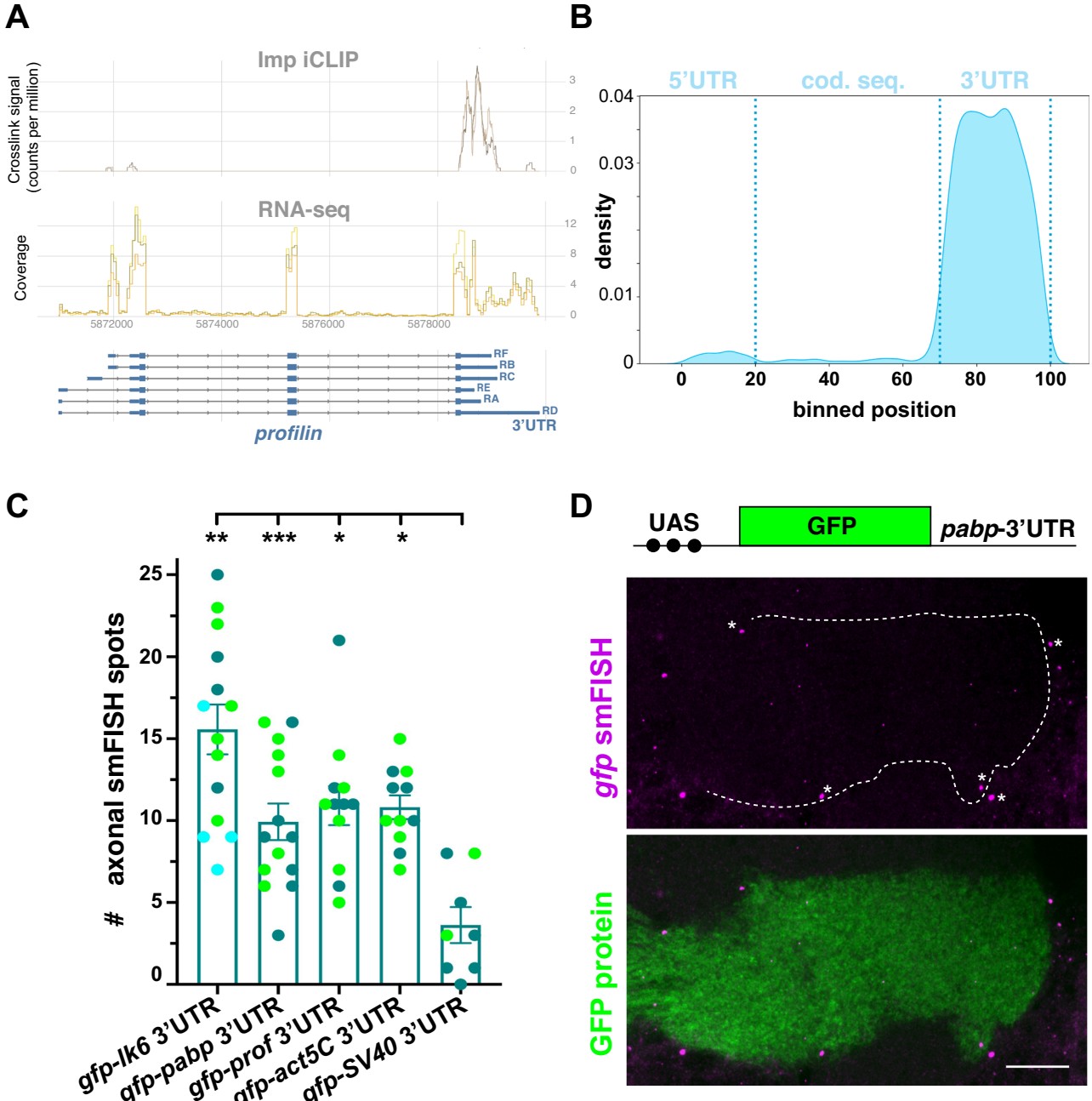

**Fig. 4 | The 3'UTR of Imp mRNA targets mediate binding and promote axonal localization. A** Profiles of the Imp iCLIP (top, two replicates) and RNA-seq (middle, three replicates) signals along the *profilin* gene regions. Profiles were generated using the *clipplotr* tool[76] and its smoothening function. Intronic and exonic sequences are represented at the bottom by single lines and boxes, respectively (large boxes for coding exons and smaller boxes for UTRs). **B** Metatranscript profile representing the distribution of Imp iCLIP peaks along a reference transcript. Significant peaks were mapped onto a virtual common transcript based on their relative position within each mRNA. **C** Numbers of *gfp* smFISH spots detected in MB γ axons after expression of UAS-driven *gfp*-3'UTR reporters with the 201Y-Gal4. *gfp*-SV40 3'UTR transcripts were used as a negative control. Two to three biological replicates were performed, and data color-coded based on the replicate they belong to. Bar graphs and error bars represent, respectively, for each mRNA, the average and SEM of all combined data points. *, $P < 0.05$; ***, $P < 0.001$ (Kruskall–Wallis with Dunn's post-tests). Exact $P$ values: 0.0097 (*gfp-lk6* 3'UTR), <0.0001 (*gfp-pabp* 3'UTR), 0.0377 (*gfp-prof* 3'UTR), 0.0113 (*gfp-act5C* 3'UTR). Numbers of brains analyzed: 11 (*gfp-lk6* 3'UTR), 14 (*gfp-pabp* 3'UTR), 14 (*gfp-prof* 3' UTR), 12 (*gfp-act5C* 3'UTR), 8 (*gfp-SV40* 3'UTR). **D** mRNA localization of *gfp-pabp* 3' UTR transcripts expressed using the 201Y-Gal4 driver. Top: schematic representation of the reporter analyzed. Middle: smFISH signal (magenta) obtained using an anti-*gfp* probe set. Asterisks point to probe aggregates. Bottom: overlay between GFP protein signal (green) and *gfp* RNA signal (magenta). Scale bar: 10 μm. Source data are provided as a Source Data file.

(Fig. S7B). These results thus indicate that the PLD of Imp is not essential for the RNA binding capacity and specificity of Imp. Together, our past and recent findings thus demonstrate that the *imp-ΔPLD* mutation disrupts the axonal transport function of Imp while impacting only to a minor extent the RNA binding capacity of Imp.

To then monitor if the localization of Imp mRNA targets to MB γ axons is altered in the *imp-ΔPLD* mutant context, we quantitatively compared the number of axonally-localized mRNAs detected by smFISH experiments in control and *imp-ΔPLD* flies. While *lk6, crc, bnb, elF4E-1* and *pabp* mRNAs did not exhibit a reproducible decrease in mRNA

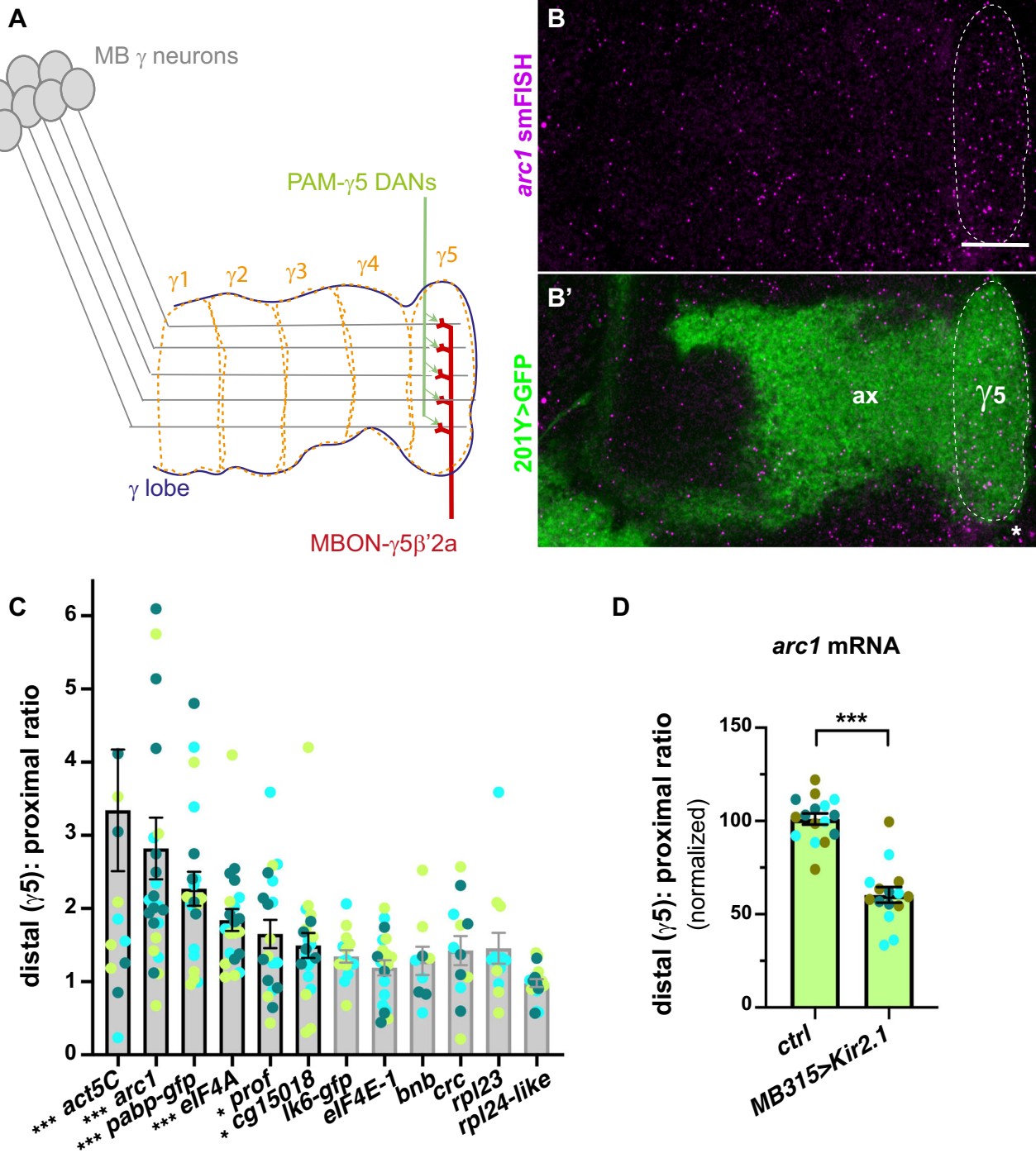

localization in the *imp-ΔPLD* context, a significant decrease in axonal localization was observed for *eIF4A*, *cg15098*, *profilin* and *actin5C* mRNAs, and a near loss of axonal signal for *arc1* mRNA (Fig. 6C–E and Supplementary Data 3). Such a decrease was not accompanied by changes in RNA levels (Fig. S8A, B), indicative of an alteration in RNA localization rather than RNA stability. Decreased axonal mRNA localization was also not observed for the non-Imp-bound mRNAs *rpl24-like* and *rpl23* (Fig. S8C), suggesting that Imp-dependent mRNA transport is selectively altered in the *imp-ΔPLD* mutant. To further confirm that the *imp-ΔPLD* mutation impairs 3′UTR-dependent mechanisms, we analyzed the axonal localization of *profilin* and *actin5C gfp*-3′UTR reporter RNAs. As shown in Fig. S8D, a significant decrease in axonal targeting was observed for both, indicating that the 3′UTR-dependent localization of these mRNAs relies on Imp-mediated axonal transport.

Together, these experiments revealed that the axonal transport of selected Imp mRNA targets is altered in *imp-ΔPLD* brains.

## Long-term, but not short-term, courtship memory is altered in *imp-ΔPLD* mutants

The fact that Imp RNA transport factor is very selectively recruited to the axons of MB γ neurons and blocked in *imp-ΔPLD* mutants provides a unique context in which to study in vivo, in the context of physiological memory circuits, the function of presynaptic mRNA targeting in the establishment of long-term memory.

To analyze the memory performance of *imp-ΔPLD* mutants, we performed courtship conditioning, in which the function of MB γ neurons was shown to be required for both short-term memory (STM) and protein synthesis-dependent long-term memory (LTM)[44,45]. In this

**Fig. 5 | Imp mRNA targets exhibit compartment-specific localization in MB γ axons. A** Schematic representation of MB axonal compartments. MB γ neurons project their axons in the medial lobe, which has been divided into 5 anatomical compartments, from proximal γ1 to distal γ5. The most distal γ5 compartment receives input from dopaminergic PAM- γ5-DAN neurons and establishes synapses with the glutamatergic output MBON- γ5β′2a neuron. **B** Confocal images of smFISH signals (magenta) obtained with anti-*arc1* probe sets. The population of MB γ axons (ax) is labeled using the 201Y-Gal4 driver and UAS-cGFP (green in **B**′). The limits of the γ5 axonal compartment are marked with a dotted white line. The asterisk indicates signals corresponding to the soma of neighboring neuronal populations. Scale bar: 10 μm. **C** mRNA enrichment in the γ5 axonal compartment for Imp-bound transcripts. Two non Imp-bound transcripts are shown as controls (*rpl23* and *rpl24-like*; right). smFISH experiments were performed in 201Y-Gal4>cGFP males. Bars with a gray border line correspond to mRNAs whose γ5 enrichment is not significantly different from that of *rpl24-like* mRNA. Bars with a black border line correspond to mRNAs whose γ5 enrichment is significantly different from that of rpl24-like (*, $P < 0.5$; ***, $P < 0.001$; Kruskall–Wallis with Dunn's post-tests). Exact $P$ values: 0.0001 (*act5C*), <0.0001 (*arc1*), <0.0001 (*pabp-gfp*), <0.0001 (*eIF4A*), 0.0163 (*prof*), 0.0454 (*cg15O18*), 0.3117 (*lk6-gfp*), >0.9999 (*eIF4E-1*), >0.9999 (*bnb*), 0.2166 (*crc*), 0.5030 (*rpl23*). Numbers of brains analyzed: 14 (*act5C*), 24 (*arc1*), 21 (*pabp-gfp*), 21 (*eIF4A*), 19 (*prof*), 21 (*cg15O18*), 15 (*lk6-gfp*), 20 (*eIF4E-1*), 9 (*bnb*), 12 (*crc*), 13 (*rpl23*), 19 (*rpl24-like*). **D** *arc1* mRNA enrichment in the γ5 axonal compartment. Precise genotypes: UAS-Kir2.1-GFP/+; MB247-dsRed/+ (control) and UAS-Kir2.1-GFP/+; MB247-dsRed/MB315C-Gal5 (MB315C>Kir2.1). ***, $P < 0.001$ (Two-tailed Mann-Whitney test). Exact $P$ value: <0.0001. In **C, D** mRNA enrichment in the γ5 compartment was quantified as the ratio of smFISH spot density (number of spots per μm³) in the distal γ5 compartment *vs* the proximal γ2-4 compartments. Three biological replicates were performed, and data points were color-coded based on the replicate they belonged to. Bar graphs and error bars represent, respectively, for each mRNA, the average and SEM of all combined data points. In **C** one and three, outliers data points were respectively omitted from the *arc1* and *act5C* graphs (but considered to calculate the mean). Numbers of brains analyzed: 16 (*ctrl*), 15 (*MB315>Kir2.1*). Source data are provided as a Source Data file.

paradigm, individual naïve males learn to suppress their courtship after being rejected by recently mated, and therefore unreceptive, females. They maintain low courtship levels towards mated females for hours (STM) or days (LTM) upon short and long training sessions, respectively[60,61]. These experiments revealed that LTM, but not STM, was strongly impaired in *imp-ΔPLD* flies when compared to controls (Fig. 7A, B). As memory acquisition and memory consolidation were previously shown to rely on the activation of the same neuronal pathway[45], this result suggests a specific role for Imp-dependent mRNA localization in memory consolidation rather than an effect on the circuit itself. Consistent with this idea, no significant differences were observed when comparing the overall density and intensity of Bruchpilot-labeled presynaptic active zones in controls *vs imp-ΔPLD* flies (Fig. S9A, B). Next, because *imp-ΔPLD* flies express mutant proteins in all *imp*-expressing cell types, we performed rescue experiments in which we re-expressed a wild-type copy of Imp in MB γ neurons using the 201Y-Gal4 driver. Defective LTM could be significantly rescued in this condition, consistent with a requirement of Imp transport function in MB γ neurons (Fig. 7C). Last, to exclude a developmental requirement of *imp* function, we inactivated Imp specifically in adult MB neurons using the Gal80ts/Gal4 inducible system to degrade endogenous GFP-Imp proteins[62,63]. While males raised at restrictive temperature (20 °C; no Gal4 activity) showed normal LTM, males shifted to permissive temperature (30 °C; Gal4 active, inducing GFP-Imp degradation) upon eclosion exhibited impaired LTM, but normal STM (Fig. S9C, D), indicating that Imp function is required specifically in adult circuits.

Together, these results thus demonstrate that the transport of Imp-dependent mRNAs to MB γ axons is required for establishment of courtship LTM. They provide evidence that axonal targeting of mRNAs is required in vivo, in the context of memory circuits, for memory consolidation.

## Discussion

Targeting of neuronal mRNAs to axonal and dendritic terminals, coupled with regulated onsite translation, has emerged as an important mechanism underlying local proteome remodeling and establishment of different forms of neuronal plasticity[2,3]. Although transcriptomics studies have revealed that a rich repertoire of mRNAs is localized to pre- and post-synaptic compartments in the mature mammalian brain[4–6], important questions remain to be addressed related to the specificity, regulation, and function of this process in the context of physiological memory circuits. To explore these questions in the genetically tractable *Drosophila* model, we first characterized the extent of mRNA localization in the adult *Drosophila* brain and profiled the RNA content of synaptosome preparations. This revealed the presence of hundreds of mRNA species encoding a variety of proteins,

but also a striking degree of conservation in the functional classes as well as in the identity of enriched mRNAs from vertebrates to invertebrates[40]. The rich dataset we have generated constitutes a valuable resource to further explore how these mRNAs are targeted to synapses and if their targeting is population-specific. Analysis of transgenic reporter constructs in which the coding sequence of GFP was fused to the 3′UTR of localized mRNAs bound by the conserved RBP Imp/ZBP1 has indicated that axonal targeting is mediated through 3′UTR-located localization elements. This is consistent both with previous knowledge on the localization function of 3′UTR sequences[22–25,64] and with the results of our iCLIP analysis pointing to a strong specificity of Imp binding to CAA-rich 3′UTR motifs. Imp binds to multiple locations along 3′UTRs in most mRNAs, therefore, further studies will be needed to assess whether a combinatorial recognition is at play or if short zipcode sequences act in isolation to determine mRNA localization. Consistent with a multifactorial model, recent massively parallel assays aimed at systematically identifying neuronal localization elements have suggested that the localization potential of most mRNA species is broadly encoded along 3′UTR length[22]. This is also in line with our discovery that a number of Imp-bound, axonally-localized mRNAs still localize in the *imp-ΔPLD* context. Those mRNAs might be less sensitive to the changes in RNP granule dynamics induced by the absence of Imp PLD[58]. Alternatively, they may be recognized by additional RBPs that function in a complementary manner to Imp to promote axonal targeting.

Whether local translation of mRNAs targeted to synapses is required to establish long-term, protein synthesis-dependent memories has remained a long-standing question. Here, the strong and specific memory deficit we have observed in *imp-ΔPLD* transport mutants demonstrated the importance of localizing specific sets of mRNAs to the presynaptic terminals of MB γ neurons for the consolidation of long-term memories. Interestingly, MB γ neurons were previously shown to be required for protein synthesis-dependent formation of long-term memory in the context of courtship conditioning, through sleep-dependent reactivation of a recurrent circuit involving the PAM γ5 DANs dopaminergic input neurons and MBON-γ5β′2a output neurons[45,65,66]. This, together with our discovery that a number of Imp-bound mRNAs accumulate within the MB γ5 compartment, raises the very interesting possibility that the activation of compartment-specific circuits promotes the local recruitment of mRNAs, thus generating compartmentalized reservoirs for local translation and both selective and long-term tagging of synapses. In this model, targeting of specific mRNA subsets to selected MB compartments may contribute to the definition of distinct functional units along axons and to the induction of local synaptic plasticity upon integration of sensory and modulatory signals[54,67–69]. Consistent with such a model, we here uncovered that the local accumulation of mRNAs in the γ5 compartment is triggered by the

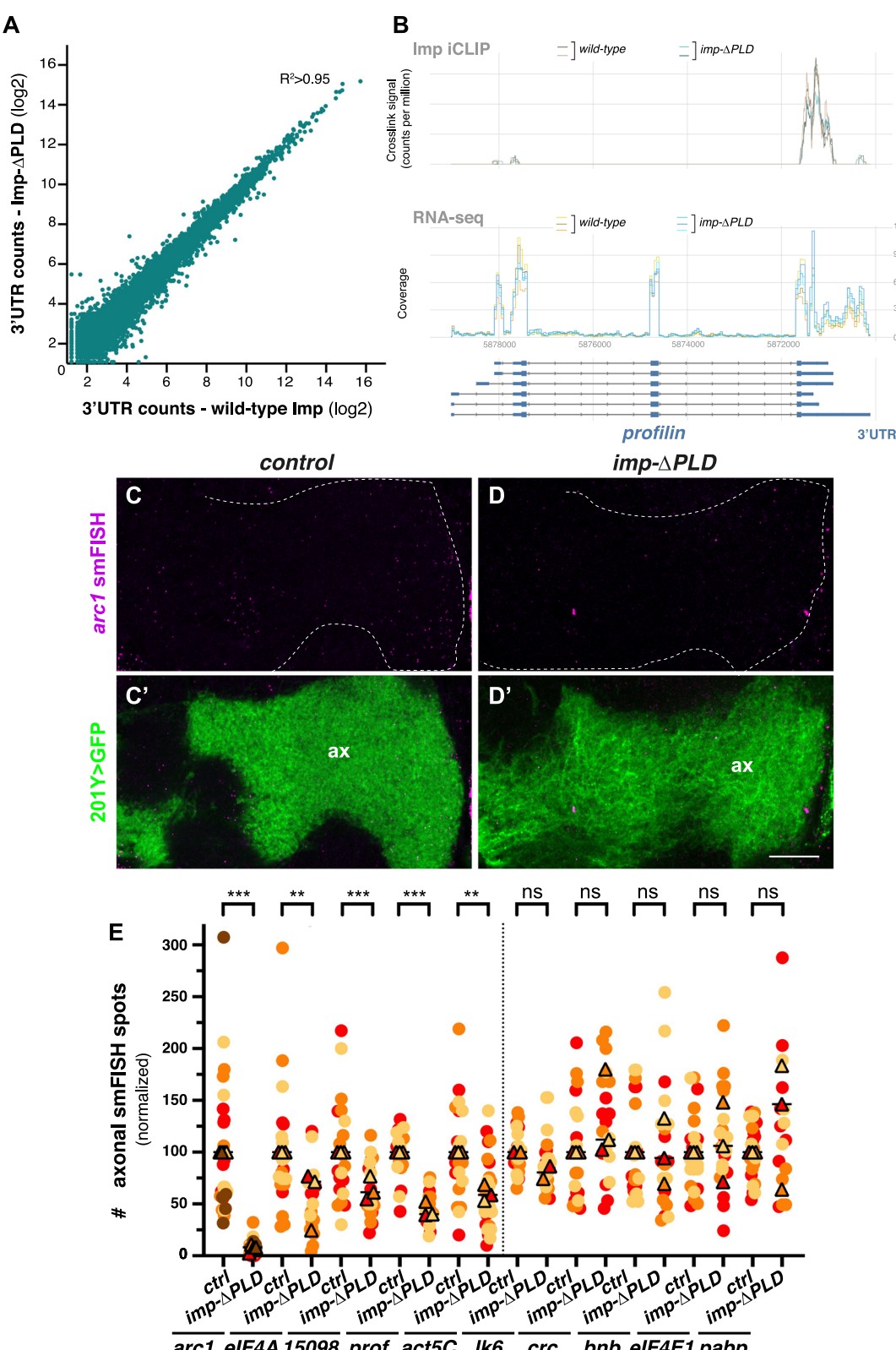

activity of upstream PAM-γ5-DANs. How specific mRNA subsets are recruited to distinct axonal sub-compartments remains to be investigated, but one could speculate that scanning mRNA molecules get locally anchored upon synaptic stimulation. Such a docking mechanism has been observed in cultured neurons where *rgs4* mRNA and *β-actin* mRNA were shown to be selectively captured by stimulated dendritic spines upon spatially resolved glutamate uncaging[11,35,36]. Synaptic

capture of *β-actin* required an intact actin cytoskeleton as well as the function of the ZBP1 RBP[35], suggesting that mRNA molecules are retained at activated synapses through *trans*-acting factors that mediate anchoring to the synaptic cytoskeletal network, a mechanism that could be at play at MB synapses.

Together, our study uncovered highly specific targeting processes that mediate neuronal mRNA localization to the presynapse of

**Fig. 6 | mRNA localization to MB γ axons, but not RNA binding, is altered in *imp-Δ PLD* mutants. A** Correlation plot showing the normalized counts of 3′UTR-mapped reads obtained for individual RNAs in wild-type (*x*-axis) and Imp-ΔPLD (*y*-axis) iCLIP experiments. **B** Profiles of the Imp wild-type and Imp-ΔPLD iCLIP signals (top, two replicates for each condition) and corresponding input RNA-seq signals (middle, three replicates for each condition) along the *profilin* gene region. Profiles were generated using the *clipplotr* tool[76] and its smoothening function. Intronic and exonic sequences are represented at the bottom by single lines and boxes, respectively (large boxes for coding exons and smaller boxes for UTRs).
**C**, **D** Confocal images of *arc1* smFISH signals (magenta) obtained in control (left) and *imp-ΔPLD* (right) contexts. The population of MB γ axons (ax) is labeled in green in **C′** and **D′** using the 201Y-Gal4 driver and UAS-cGFP. Scale bar: 10 μm.
**E** Normalized numbers of smFISH spots detected in MB γ axons of wild-type and

*imp-ΔPLD* mutants. Two to four biological replicates were performed, and the mean value of each is indicated as a triangle. Data points were color-coded based on the replicate they belonged to. 15098 stands for *cg15098*. **, *P* < 0.01; ***, *P* < 0.001 (Two-tailed Mann–Whitney tests). n.s. stands for not significant. Two outlier data points (one for *arc1* and one for *pabp*) were omitted from the graph (but considered to calculate the mean and to perform the statistical tests). Exact *P* values: <0.0001 (*arc1*), 0.0058 (*eIF4A*), 0.0008 (*cg15098*), <0.0001 (*prof*), 0.0035 (*act5C*), 0.0719 (*lk6*), 0.0902 (*crc*), 0.9330 (*bnb*), 0.4527 (*eIF4-E1*), 0.1925 (*pabp*). Numbers of brains analyzed for control and *imp-ΔPLD* conditions respectively: 24 and 25 (*arc1*), 21 and 19 (*eIF4A*), 21 and 20 (*cg15098*), 19 and 20 (*prof*), 20 and 25 (*act5C*), 24 and 22 (*lk6*), 18 and 19 (*crc*), 22 and 19 (*bnb*), 22 and 22 (*eIF4-E1*), 22 and 17 (*pabp*). Source data are provided as a Source Data file.

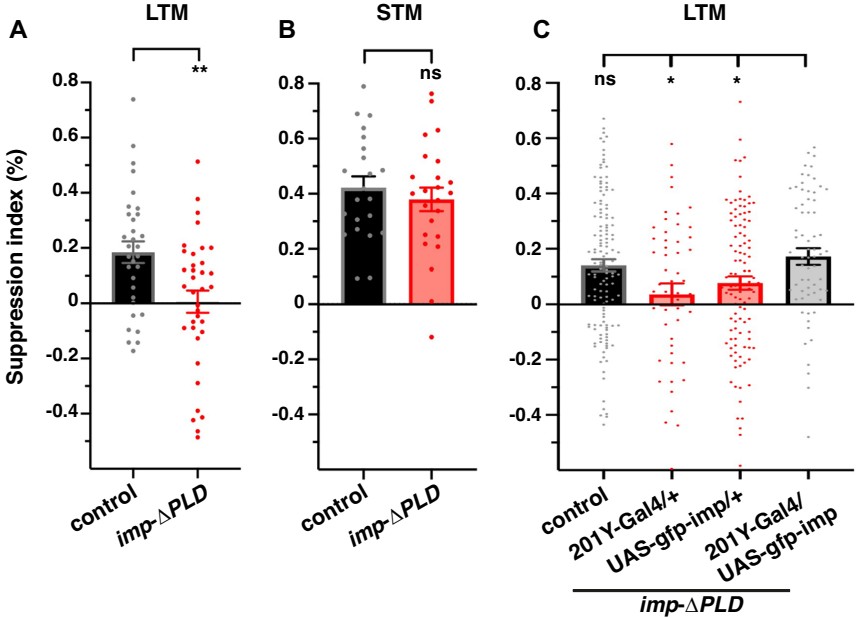

**Fig. 7 | *imp-ΔPLD* males exhibit defective long-term courtship suppression.**
Courtship suppression indices reflecting the Long-Term Memory (LTM, **A**) and Short-Term Memory (STM, **B**) performances of controls (black) and *imp-ΔPLD* (red) mutants. Number of trained males tested for each genotype: *Canton S (controls)*: *n* = 32 (LTM) and 22 (STM); *imp-ΔPLD*: *n* = 36 (LTM) and 24 (STM). **, *P* < 0.01 (Two-tailed unpaired *t*-tests). n.s. stands for not significant. Exact *P* values: 0.0024 (LTM), 0.4713 (STM). **C** Long-term courtship suppression indices of *imp-ΔPLD* mutants (red) and mutants in which a wild-type copy of *imp* has been re-expressed in MB γ

neurons (gray). *Canton S (controls)*: *n* = 132; *imp-ΔPLD; 201Y-Gal4/+*: *n* = 52; *imp-ΔPLD; UAS-gfp-imp/+*: *n* = 108; *imp-ΔPLD; 201Y-Gal4/UAS-gfp-imp (rescue)*: *n* = 67. *P* < 0.05 (One-way ANOVA test followed by Dunnett's multiple comparison tests). n.s. stands for not significant. Exact *P* values: 0.7706 (*controls*), 0.0389 (*imp-ΔPLD; 201Y-Gal4/+*), 0.1547 (*imp-ΔPLD; UAS-gfp-imp/+*). Data points represent individual males, bar graph mean values, and error bars SEM. Source data are provided as a Source Data file.

memory neurons and demonstrated their functional requirement in vivo. It also paved the way for the exploration of how circuit activity regulates mRNA recruitment and downstream translation in the functional context of memory consolidation.

## Methods

### *Drosophila melanogaster* stocks and genetics
Flies were raised on standard media at 25 °C and dissected 4–8 days post-eclosion. While males and females were used for experiments shown in Figs. 2, 4, 7, S2 and S3, males were exclusively used for experiments shown in Figs. 3, 5, 6, and S6C. The following fly stocks were used for smFISH experiments: Imp-ΔPLD (in which the GFP cassette from the original G080 line has been excised through P-element mobilization)[58]; VT44966-Gal4 (VDRC stock center), 201Y-Gal4; 201Y-Gal4,UAS-cGFP; 201Y-Gal4,UAS-CD8-RFP; UAS-*gfp-profilin*-3′UTR and UAS-*gfp-SV40* 3′UTR[70]; UAS-CD8-GFP;;OK107-Gal4; UAS-*profilin*-RNAi (BDSC#34523); *lk6-gfp* (BDSC#59795), *csas-gfp* (BDSC#67737), *rols-gfp* (BDSC#64471) and *pabp-gfp*[71] protein-trap or knock-in lines. *elav*-Gal4, UAS-Syt-1-eGFP (BDSC#6923) flies were used for the western blot

shown in Fig. 1B. The MB315C-Gal4 (BDSC#68316), UAS-Kir2.1-GFP (BDSC#6596), and MB247-dsRed (gift from T. Riemensperger) were used for respectively silencing PAM-γ5 DAN neurons and visualizing MB neurons. The following stocks were used for courtship experiments: Canton S (gift from Krystyna Keleman), cantonized imp-ΔPLD flies; UAS-*gfp-imp*[47]; 201Y-Gal4.

### Generation of *gfp*-3′UTR reporter lines
The UASp-EGFP-3′UTR constructs were generated by LR recombination using pENTR:D/TOPO donor plasmids containing 3′UTR sequences and a UASp-EGFP-W destination vector[70]. The 3′UTR sequences were PCR-amplified using the following primers: *pabp*_fwd (5′- CACC GCTCGAACAGC TCAAGCGTATG -3′) and *pabp*_rev (5′- ATAGATATT AAACATAAAAATCCATCC -3′); *lk6*_fwd (5′-CACCGCGGGTCCACTGTGG ACAGATAAC -3′) and *lk6*_rev (5′- CATGTATTTAGTGTTTTT ATTGA G-3′); *act5C*_fwd (5′- CACCGAAGGATCGCTTGTCTGG -3′) and *act5C*_rev (5′-TGTTGTTGTTTCATTTCATCAG-3′); *arc1*_fwd (5′ – CACCGCGACA AAAAGAACATCAAATACC – 3′) and *arc1*_rev (5′- CCGTTTCTGAGTTT AATG GTTG - 3′).

## Synaptosome preparation

Synaptosome fractionation was adapted from ref. 39. Four hundred miligram of frozen *Drosophila* adult heads were homogenized in 7 volumes (2.8 mL) of homogenization buffer 0.32 M sucrose, 20 mM HEPES−KOH pH 7.4 supplemented with protease and phosphatase inhibitor cocktail (complete EDTA-free) and vanadyl-ribonucleoside complex 1X (VRC) in a pre-chilled 15 mL dounce type glass homogenizer, through around 40 manual gentle strokes, avoiding air bubbles. The resulting head homogenate (H) was then transferred into 2 mL tubes and centrifuged twice at $1000 \times g$ at 4 °C for 10 min to remove nuclei and debris that pellet in the P1 fraction. S1 supernatants were then centrifuged again 20 min at $16,000 \times g$ at 4 °C to separate the cytoplasmic material (supernatant S2) from synaptosomes and mitochondria (pellet P2). P2 pellets were resuspended together in 1.5 mL of homogenization buffer with an eroded pasteur pipette and loaded on top of a 10 mL discontinuous sucrose gradient composed of 5 mL of a 0.8 M sucrose phase (0.8 M sucrose, 20 mM HEPES−KOH pH7.4, VRC 1X) layered on top of 5 mL of a 1.2 M sucrose phase (1.2 M sucrose, 20 mM HEPES−KOH pH7.4, VRC 1X). The sucrose gradient was centrifuged at $54,000 \times g$ for 1 h 30 min at 4 °C in a SW-41-Ti swinging-bucket rotor. The synaptosome fraction was collected with an eroded pasteur pipette at the interface between the 0.8 M sucrose and 1.2 M sucrose phases, washed with 9 mL of homogenization buffer without VRC, and centrifuged at $20,000 \times g$ for 30 min at 4 °C. The final pellet (Synapt) was resuspended in 100 μL of homogenization buffer without VRC and then processed for RNA extraction. One hundred microliters of the H fraction were used in parallel for RNA extraction.

## Electron microscopy on synaptosome fraction

The synaptosome fraction was pelleted and fixed for 1 h in 2.5% glutaraldehyde in homogenization buffer. The fixation solution was then removed and replaced by 0.5% glutaraldehyde in Hepes buffer, and the sample was stored at 4 °C. The synaptosome fraction was then rinsed in PHEM buffer and post-fixed in a 0.5% osmic acid +0.8% potassium Hexacyanoferrate trihydrate for 2 h in the dark at room temperature. After two washes in PHEM buffer, the synaptosome fraction was dehydrated in a graded series of ethanol solutions (30–100%) and embedded in EmBed 812 using an Automated Microwave Tissue Processor for Electronic Microscopy (Leica EM AMW). Thin sections (70 nm; Leica-Reichert Ultracut E) were collected, and the sections were counterstained with uranyl acetate 1.5% in 70% Ethanol and lead citrate and imaged using a Tecnai F20 transmission electron microscope at 120 KV.

## Synaptosome RNA-seq and analysis

Synaptosomes were prepared in triplicates from GFP-Imp#G080 flies, and RNA was extracted from both head homogenates (H, input) and synaptosome fractions. The quality of total RNA was checked on a picoRNA chip with a bioanalyzer, and rRNA was depleted using the QIAseq FastSelect −rRNA Fly Kit (ID# 333262), followed by library preparation using the NEBNext RNA Ultra II Library Prep Kit. Profiles of the corresponding libraries were checked on a HS DNA screen tape with a Tapestation 4150, multiplexed and 2*50 bp paired-end sequenced on an Illumina NextSeq device. Between 65.7 and 120.1 million reads were obtained.

Reads were processed using custom bash scripts. First, we used BBsplit from BBTools (39.01) (http://sourceforge.net/projects/bbmap/) to eliminate reads with ambiguous mapping. Next, we trimmed reads to remove adapters and low-quality base calls with FASTP (0.22.0)[72], thus keeping from 11.2 to 43.3 million reads of 30–31 bp per sample. These reads were aligned to the *Drosophila melanogaster* genome (dm6) with STAR (2.7.10a)[73] in paired-end mode, using BDGP6.32.107 assembly as gene annotation from ENSEMBL. More than 84.2 % of reads were thus successfully mapped in the different replicates (uniquely mapped + mapped to many loci), imported to R (4.2.0), and

counted using featureCounts from the Rsubread package (2.12.0). For DGE analysis, FBti** and RR** sequences were excluded because of mapping ambiguities. Genes that had no count in all samples or did not have a CPM > 0.5 in at least 3 samples were also removed. Input and synaptosome samples coming from the same initial lysate were paired to perform DGE using DESeq2 (1.38.0)[74] with alpha = 0.05 and pAdjustMethod = "BH" and the synaptosome samples as the reference levels. Only genes with an adjusted *P* value < 0.05 were considered for further analysis.

Gene Ontology was performed with Gorilla (http://cbl-gorilla.cs.technion.ac.il) on the synaptically-enriched genes (log2FC ≥ 0.85), using the remaining significant genes of our dataset (log2FC ≤ 0.85) as background. The GO terms returned by Gorilla may sometimes exhibit redundancy. To overcome this, we calculated a jacquard index to cluster GO terms based on the proportion of shared genes (Fig. S1C). This similarity index was calculated as the ratio between the intersection and union of two GO classes (termed A and B):

$$d(A,B) = 1 - \frac{|A \cap B|}{|A \cup B|}$$

## Western blot

10–20 μg of the head lysate (H), P1, and Synaptosome soluble and insoluble protein fractions were used for Western-Blotting. The latter fractions were obtained after lysis of the synaptosome fraction in 2 % Triton, 10 mM Tris-HCl and complete EDTA-free 1X (Roche, # 11873580001), incubation for 5 min on ice and centrifugation for 10 min at $16,000 \times g$ at 4 °C. Soluble synaptic proteins were recovered from the supernatant while insoluble proteins were recovered after resuspension of the pellet with homogenization buffer. Protein concentrations were determined for each fraction using the Bradford method, and samples were loaded on NuPAGE 4–12 % Bis-Tris precasted gels for migration and then transferred to nitrocellulose membranes. After blocking, membranes were incubated with different primary antibodies (mouse anti-Bruchpilot (NC82 (DSHB); 1:1000)), mouse anti-Lamin (Dm0 67.10 and 84.12 (DSHB); 1:2000 each), mouse anti-Cystein String Protein (Ab49 (DSHB); 1:200), mouse anti-Elav (7E8A10 (DSHB); 1:500 or rabbit anti-GFP) (#TP401 (Torey Pines); 1:000) overnight at 4 °C with agitation. Membranes were then washed with PBS Tween 0.1% and incubated with fluorescent secondary antibodies (goat anti-rabbit AF680 (Invitrogen, #A21076, 1:10,000)), goat anti-mouse IRDye 800 (Invitrogen, #SA-10156, 1:10,000) for 2 h at room temperature. After three washes with PBS Tween 0.1%, the fluorescence was detected using an Odyssey LiCoR system. Blot uncropped scans are provided in the Source Data file.

## Single molecule fluorescent in situ hybridization (smFISH)

Brains from 5 to 7-day-old *Drosophila* were dissected in cold RNase-free HL3 buffer. Dissected brains were then fixed in 4% formaldehyde in HL3 buffer for 1 h at 4 °C, rinsed twice with PBS, and stored overnight at 4 °C in 70% Ethanol in PBS. On the next day, brains were treated with Proteinase K (#AM2546; 2 μg/mL) in 2x SSC for 5 min at RT and then washed twice with PBS, followed by wash buffer (10% formamide in 2x SSC) for 5 min.

For smFISH experiments performed with stellaris probe sets, brains were then incubated overnight, at 45 °C, and under agitation, with Quasar®570/670- labeled Stellaris® Probes in 100 μL hybridization buffer (100 mg/mL dextran sulfate, 10% formamide in 2x SSC). e*gfp and camk2* probes were used at a final concentration of 0.125 μM and *profilin* at a concentration of 0.25 μM. *profilin* and *gfp* probes used were identical to those used in refs. 70,71, respectively, and can be found in Supplementary Data 7. Sequences

of the *camk2* and *actin5C* probes can be found in Supplementary Data 7.

For smFISH experiments performed using the smiFISH approach[46], brains were then incubated overnight at 37 °C, under agitation with 1.25 µL of a probe duplex stock solution for each 50 µL of hybridization buffer (Stellaris RNA hybridization, 10% formamide). Probe duplex stock solutions were pre-prepared by mixing individual primary probes (sequences in Supplementary Data 7) with complementary Cy3-FLAPx-Cy3 or Atto647-FLAPx-Atto647 (Eurofins genomics) in TE-NaCl 100mM buffer to reach a final concentration of 0.05 µM/primary probe and 5 µM, respectively. Probe duplex stock solutions were then heated to 95 °C for 5 min, cooled down until 35–37 °C, and incubated in ice for 30 min. Sequences of the smiFISH probes can be found in Supplementary Data 7.

After hybridization, brains were washed twice for 30 min in pre-warmed wash buffer under agitation at 37 or 45 °C and mounted in Vectashield (Vector Laboratories) medium.

## Image acquisition
Brain samples were imaged using an LSM880 confocal equipped with an Airyscan module and a 63 × 1.4 NA oil objective. Images were taken with a 0.07 µm xy pixel size and a 0.25 µm z step size and processed with the automatic Airyscan processing module of Zen.

## smFISH signal quantification
**smFISH signals in cell bodies.** ROIs (300 × 300 pixels) were cropped from single z slice,s and image intensities rescaled to enhance contrast and keep 0.01% pixels saturated. smFISH spots were detected using the Small Particle Detection (SPaDe) algorithm (https://raweb.inria.fr/rapportsactivite/RA2016/morpheme/uid13.html)[75]. The cutoff size for smFISH spots was set to 4 pixels, and the threshold used for the detection of smFISH spots was 0.62, except for *rpl24-like* (0.42), *eIF4A* (0.32), and *bnb* (0.82).

**smFISH signals in axons.** Z-stack images containing 25 slices covering a ~6 µm thick section of MB axonal lobes labeled with GFP or RFP were cropped to standardize images and include exclusively compartments γ2-5. A pipeline was developed on the Imaris software ensuing the following steps: (1) selection of the ROI (γ2-5 compartments) based on lobe absolute fluorescent intensity, using surface function with automatic thresholding and surface detail of 0.146 µm; (2) detection of smFISH spots using the spot tool, with estimated XY diameter of 0.4 µm and PSF-elongation along *Z*-axis of 0.8 µm; (3) overlapping of the two signals, using the "spots close to surface" filter to include only spots inside the γ lobe (shortest distance to surface <0), with a further selection of spots comprised within the distal 2/3 of the lobe (position measured along the medial lobe axis) (Fig. S10). For proximal:distal ratio, MB γ lobe volume was divided into two parts: the γ5 compartment and the remaining 2/3 of the lobe, and the number of spots in each region was divided by the corresponding volume. Samples where the number of axonal smFISH spots was lower than 10 were excluded from the analysis.

## Immunostaining on whole-mount adult brains
Brains were dissected in cold HL3 buffer for 1 h and fixed in 4% formaldehyde, HL3 buffer for 1 h. After three washes in 0.1% PBS/ Triton-X (PBT), brains were blocked overnight in PBT supplemented with 1% BSA. The next day, brains were incubated with mouse α-NC82 primary antibody (DSHB, 1:100) for 24 h. Brains were then washed thrice in PBT 0.1% and incubated with α-mouse secondary antibodies conjugated with Alexa Fluor 568 (Thermo Fisher, 1:500) overnight at 4 °C. Brains were washed thrice in PBT 0.1% and mounted in Vectashield (Vector Laboratories) medium.

## Imp iCLIP and analysis
Samples were prepared in replicates from GFP-Imp#G080 flies and GFP-Imp-ΔPLD adult heads. Frozen fly heads were kept on dry ice, and porcelain pestles and mortars were pre-cooled on dry ice. Fly heads were ground until a fine powder was generated. Powder was transferred to a 6-well tissue culture (TC) plate that was pre-cooled on dry ice. Samples were irradiated 4x with 150 mJ/cm² in a Stratalinker 2400 at 254 nm while still on dry ice. In between x-linking, the powder is mixed to guarantee homogenous cross-linking. Tissue powder was lysed in the TC dish after moving it onto normal ice in RIPA buffer (with Protease inhibitor). 0.2 Units of RNaseI and 4 Units Turbo DNase were added per 1 mL of cell lysate at 1 mg/mL protein concentration for RNA fragmentation. GFP-Trap® MA beads (ABIN1889489, ChromoTek) were prepared according to the supplier's instructions and used to isolate Protein-RNA complexes. RNA was ligated to a pre-adenylated infrared labeled IRL3 adaptor with the following sequence:*/5rApp/AG ATC GGA AGA GCG GTT CAG AAA AAA AAA AAA /iAzideN/AA AAA AAA AAA A/3Bio/*. The complexes were then size-separated by SDS-PAGE, blotted onto nitrocellulose, and visualized by Odyssey scanning. RNA was released from the membrane by proteinase K digestion and recovered by pre cipitation. cDNA was synthesized with Superscript IV Reverse Transcriptase (Life Technologies) and AMPure XP beads purification (Beckman-Coulter, USA), then circularized using Circligase II (Epicenter) followed by AMPure XP beads purification. After PCR amplification, libraries were size-selected with Ampure beads and gel-purification and quality controlled for sequencing. Libraries were sequenced as single-end 100 bp reads on Illumina HiSeq 4000, producing more than 9.5 million reads per replicate.

**Data analysis.** iCLIP reads were demultiplexed on iMAPS (https://imaps.goodwright.com/) using iCount *demultiplex* (https://github.com/tomazc/iCount), which also moved UMIs to read headers and trimmed the Illumina 3' sequencing adapter. First, we used BBduk from BBTools to select reads with a minimum 15 bp, remove adapters from both read ends (allowing 1 mismatch), and further trim low-quality base calls in the 3' end. Next, we aligned the remaining sequences to the *Drosophila melanogaster* genome (dm6) with STAR (2.7.10a) in single-end mode and using BDGP6.32.107 assembly as gene annotation from ENSEMBL with settings --outSAMtype BAM SortedByCoordinate --quantMode GeneCounts --outFilterScoreMinOverLread 0.80. 56% (wild-type Imp, replicate 1) and 67% (wild-type Imp, replicate 2) reads were uniquely mapped. The resulting BAM files were subsequently processed to deduplication using umi_tools dedup from UMI tools with settings --edit-distance-threshold = 1, thus reducing the total number of reads from 3,257,211 to 634,505 reads for replicate 1 and from 4,912,260 to 960,067 reads for replicate 2. Counting cross-link events and peak calling were performed with iCount (https://icount.readthedocs.io/en/latest/tutorial.html" \l "quantifying-cross-linked-sites). Specifically, iCount *xlsites* was first run on each replicate with settings --group_by start --quant cDNA, which gave us identified and quantified crosslinked sites. After the counting, we pooled crosslinked sites from both replicates using iCount *group*. Last, we used iCount *peaks* to generate a subset of significantly cross-linked sites with the following parameters: *iCount peaks regions.gtf cDNA_unique.bed sample_regions_peaks_unique.bed --scores sample_regions_scores_unique.tsv. regions.gtf is a specific annotation file returned by iCount segment with the following settings: iCount segment Drosophila_melanogaster.BDGP6.32.107.gtf Drosophila_melanogaster.BDGP6.32.dna.toplevel.fa.fai.*

Motifs enriched around significant peaks were identified with PEKA[76] using iCount xlsites and iCount peaks outputs and the *Drosophila_melanogaster.BDGP6.32.dna.toplevel.fa.fai.* Motif enrichment

profiles in 3′UTRs were plotted using cv_coverage.py (https://github.com/ulelab/cv_coverage/blob/main/cv_coverage.py).

**Differential gene expression for iCLIP data.** To compare the cross-linking profiles of wild-type Imp and Imp-ΔPLD, we only counted reads mapping to the 3′UTR, using featureCounts from the Rsubread package (2.12.0). Read counts of each group (WT and ΔPLD) were separately filtered as described for synaptosome RNA-seq, then normalized using DESeq2 after merging all groups. The mean of replicates was plotted for each condition.

**Metatranscript analysis.** To construct a standardized mRNA reference library from the genes recovered in the iCLIP datasets, we used the UCSC database to retrieve a ncbiRefSeq table from *Drosophila melanogaster* dm6 genome release (BDGP Release 6 + ISO1 MT/dm6). All transcripts that were not assigned to the X, Y, MT, 2L, 2R, 3L, or 3R chromosomes were first removed. Then, we kept only the longest transcript of each gene and removed all transcripts with ambiguous annotation (incomplete, unknown) and non-coding RNA. To calculate mRNA length, we summed exon lengths and converted genomic CDS start/end positions into corresponding transcript positions. 5′UTR was defined as the region between mRNA start and CDS start and 3′UTRs as the regions between CDS end and mRNA end.

For transcripts with ≥20 significant peaks, peak genomic positions were converted into corresponding transcript positions and labeled based on the transcript region into which they fall. For each gene, peak transcript positions within 5′UTR, CDS, and 3′UTR were respectively rescaled from 0 to 19, 20 to 69, and 70 to 100. The density of these rescaled positions was then computed using kernel density estimation (KDE), reflecting for each position along the transcript the overall likelihood of finding an IMP crosslinked site.

## RIP-chip and analysis

**Immunoprecipitation and RNA extraction.** 201Y-Gal4/+, UAS-Flag-Imp/+, and control 201Y-Gal4/+ flies were obtained by crosses raised at 25 °C. 3–5-day-old flies were collected and snap frozen. Heads (1.8 mL per condition) were collected at 4 °C using two prechilled sieves of different mesh sizes (630 μm on top and 400 μm at the bottom) and homogenized in a prechilled 15 mL glass Dounce Tissue Grinder with 5 mL lysis buffer (20 mM Tris-HCl-pH 8, 150 mM NaCl, 10 mM EDTA-pH 8, 0.02 mg/mL heparin, 0.2% NP40, 1.5 mM dithiothreitol (DTT), complete EDTA-free 1X (Roche, # 11873580001)) supplemented with DNAseI (20 μ/mL) and RNAse inhibitors (100 μl/mL SUPERase.In (ThermoFisher, #AM2694)). The homogenates were cleared by two consecutive centrifugations, a first at 1150 × *g* for 10 min at 4 °C and a second at 9400 × *g* for 10 min at 4 °C, and pre-adsorbed against protein-agarose beads (250 μL per condition) for 30 min at 4 °C. In parallel, mouse anti-Flag antibodies (M2 clone, Sigma; 35 μg/condition) were coupled to protein G-agarose beads (500 μL per condition) for 30 min at room-temperature, washed three times in lysis buffer, added to the pre-adsorbed lysates and incubated with the lysates on a rotator for 1.5 h at 4 °C. Beads were then pelleted by mild centrifugation (8 × *g* for 2 min at 4 °C), washed three times 15 min with lysis buffer and incubated for 30 min with proteinase K (ThermoFisher, #AM2546), first at 30 °C and then at 50 °C (10 min). Eluates were then collected and RNA recovered through Trizol extraction. Two biological replicates were performed.

RNAs were reverse-transcribed using oligodT primers, amplified, fragmented, and biotinylated following the Affymetrix GeneChip Expression Analysis Technical Manual. Labeled cDNAs were then hybridized on GeneChip *Drosophila* Genome 2.0 arrays (Affymetrix Inc., Santa Clara, CA, USA), which include 18,952 probes targeting 13,227 different genes.

**Data analysis.** Raw data were uploaded onto the Genespring software and filtered so as to eliminate gene IDs exhibiting a very low signal (<20%) in at least one condition. Normalization was performed by Robust Multi-array Analysis (RMA), without baseline transformation. Normalized signals in the bound fraction of each 201Y-Gal4/+, UAS-Flag-Imp/+ sample was compared to signals in the bound fraction of each 201Y-Gal4/+ controls and transcripts exhibiting a log2FC > 1 for each comparison were selected to produce a robust list of Imp-associated mRNAs.

## Courtship conditioning

All experiments were performed with Imp-ΔPLD flies cantonized for 5 generations and raised at 25 °C with a 12 h/12 h light/dark cycle. Trainings and tests were performed in a dedicated room where temperature was kept at 23–25 °C and humidity at 60–80%. Virgin males were collected between 0 and 4 h after eclosion and transferred to individual glass food vials, where they were aged for 5 days before space training with pre-mated females. Canton S virgin females were collected in parallel and kept in normal food vials in groups of 10. Sixteen hours before the start of training, females were pre-mated with >5-day-old Canton S males previously housed in groups of 15. For training, individual males were placed in individual small (16 × 100 mm) glass food vials and consecutively exposed to three different mated females for 2 h each, with a resting interval of 30 min. Naïve males were prepared in parallel but not exposed to any female. Females were removed from the glass vials after the last round of training, and males were kept in isolation before the test. STM was assessed 30 min after training and LTM 24 h after training.

Courtship behaviors were recorded for 12.5 min in 25 mm diameter chambers, and courtship indices (percentage of time spent by males on courting) were automatically extracted from $t = 2.5$ min onwards, using a custom-built Fiji algorithm[77]. Memory Indices (or courtship suppression indices) were calculated for each tested male as follows: $1 - (CI_{Trained} / CI_{Naïve})$, where $CI_{Trained}$ represents the courtship index of the trained fly, and $CI_{Naïve}$ represents the mean courtship index of the naive flies, respectively.

For the degradFP experiments described in Fig. S7, flies were raised at 20 °C. Half of the progeny was shifted to 30 °C upon eclosion, trained, and tested at this temperature. The other half was maintained at 20 °C until testing. In this experiment, males underwent a single round of training (1 h for STM and 6 h for LTM[65]) and were tested in 10 mm diameter chambers.

## Data availability

Materials generated for the study are available from the corresponding author on request. Source data are provided with this paper. The synaptosome RNA-seq data generated in this study have been deposited in the NCBI GEO database under accession code PRJNA1064379. The Imp iCLIP data generated in this study have been deposited in the NCBI GEO database under accession code PRJNA1063549. Source data are provided with this paper.

## Code availability

All original code has been deposited at GitHub and is publicly available under the following links: https://github.com/HibaLaghrissi/Synaptosome-and-IMP_iCLIP-analysis.git or https://doi.org/10.5281/zenodo.14921462.

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

## Acknowledgements

This study was supported by the CNRS, as well as grants from the ANR (ANR-20-CE16-0010) and the Fondation pour la Recherche Médicale (Equipe FRM; grant #DEQ20180339161) to F.B. B.dQ received fellowships from the French Ministry of Research and the Fondation pour la Recherche Médicale (FDT202204014805). This work was supported by the UCA JEDI Investments in the Future project managed by the National Research Agency (ANR) under reference number ANR-15-IDEX-01, by Howard Hughes Medical Institute at Janelia Research Campus, by the Wellcome Trust (215593/Z/19/Z), the MND association (Hallegger/Oct15/959-799), by the UK Dementia Research Institute [award number UK DRI-RE21605] through UK DRI Ltd, principally funded by the UK Medical Research Council, and by the Francis Crick Institute which receives its core funding from Cancer Research UK (CC0102), the UK Medical Research Council (CC0102), and the Wellcome Trust (CC0102). The authors are grateful to the OPAL infrastructure and the Université Côte d'Azur Center for High-Performance Computing for providing resources and support. We thank the iBV PRISM Imaging facility for use of their microscopes and support (especially B. Monterroso and S. Ben Aicha), the iBV bioinformatics platform (L. Martin and A. Fortuné) and K. Kuret for discussion and advice, the EMBL GeneCore Facility for GeneChip analysis, and the UMS2008/US40 IBSLor platform (especially V. Marchand and I. Motorine) for RNA-sequencing and guidance. We are grateful to N. Formicola for her initial input on the project, R. Barajas-Azpeleta and K. Si for their help in setting up courtship conditioning assays, and to the Besse group for discussion and advice throughout the project. We thank C. Medioni and F. De Graeve for their critical reading of the manuscript. We are grateful to the Bloomington *Drosophila* Stock Center and the Developmental Studies Hybridoma Bank for reagents.

## Author contributions

B.R.d.Q., S.R., L.B., and F.D.G. performed and analyzed the smFISH experiments. B.R.d.Q. performed the courtship conditioning experiments shown in Fig. 7. M.d.d.S. provided technical help for breeding and selection of individuals. L.B. and M.D. optimized the synaptosome fractionation and performed the blots. M.H. performed the iCLiP experiments. U.D. performed the courtship experiments shown in Fig. S9C, and D.C.C. performed the EM analysis on synaptosome fraction. H.L. performed the bioinformatic analysis of the synaptosome RNA-seq and the CLiP, under the co-supervision of F.B., A.H., and J.U. M. R. performed the bioinformatics analysis of the RIP-chip experiment. F.B., J.U., and K.K. raised funds. F.B. provided the overall supervision and wrote the original draft of the article. All authors edited and commented on the manuscript.

 

## Competing interests

The authors declare no competing interests.
