## [Peer Review file · Nature Communications]

Axonal RNA localization is essential for long-term memory

Corresponding Author: Dr Florence Besse

Version 0:

Reviewer comments:

Reviewer #1

(Remarks to the Author)

In the interesting and timely manuscript entitled Axonal RNA localization is essential for long-term memory, de Queiroz et al. identify mRNA targets in synaptosomes of *Drosophila* and confirm selected transcript localization in MB axons. Moreover, the authors identify a subset of mRNAs localized by their 3'-UTR via the Imp RNA-binding protein. Disruption of Imp transport resulted in mRNA mislocalization and deficiencies in LTM during male *Drosophila* courtship conditioning. Overall, the data is presented understandably and in a conclusive manner. The experiments are explained well and interpreted rigorously. The results expand nicely on previous research of the Besse lab and add valuable new insight to the community. Enclosed are suggestions that might expand the impact of this interesting and original study.

Main comments

- Is the compartment-specific y5 mRNA accumulation (Fig. 5) also true for act5c mRNA? While profilin mRNA accumulates in this compartment, this is not clearly seen in Fig. 4D. Could it be that the 3'-UTR is sufficient for axonal, but not compartment-specific localization?
- The finding that localized mRNAs may play a role in local circuits (Fig. 5) is fascinating, but not extensively explored in this study. Though a difficult experiment, it would be interesting to see whether the local circuitry indeed is responsible for axonal mRNA localization. Would it be possible to interfere/inhibit this circuit in brains? Or would the circuit functionality be affected by mRNA mislocalization (e.g. by mutated Imp)? Alternatively, would the protein product of localized mRNA be relevant for the circuit and is it specifically localized in y5 as well?
- As the total mRNA levels are not altered in PLD mutants, but axonal localization is disrupted for some targets (Fig. 6C-E), would mRNA accumulate in the cell body or other compartments instead? Could this be seen in FISH images?
- It would be interesting to investigate how RNA mislocalization affects courtship behavior. Is translation affected as well? Is the protein product mislocalized? Is the local circuit or are synapses altered (e.g. fewer functional synapses by immunostaining)?

Minor comments

- Please state for all graphs what the error bars represent in the figure legends (e.g. Fig. 2F, Fig. 7).
- How does the axonal localization of mRNAs presented in Fig. 3B compare to their abundance in the cell body (as done in Fig. 2F)? For easier comparison, could the authors add the means to Fig. 3B. Out of interest, how strongly were the 13 overlapping targets enriched in synaptosomes?
- In some images, it appears that mRNA is quite abundant in areas adjacent to the axonal compartment (e.g. Fig. 3). Though MB axons are clearly the focus of this study, could the authors comment on the surrounding mRNA localization, if relevant? Are these adjacent areas anatomically connected/related to the axonal compartment? By the way, the scale bar is missing in Fig. 3.
- For the data in Fig. 4E, could the authors provide example images for all conditions (main or Suppl figure)?
- As Imp iClip sites are enriched in 3'-UTRs (Fig. 4), is this true for all 13 overlapping synaptosomal targets presented in Fig. 3A, or is there variation?

(Remarks on code availability)

I am not the right person for that.

Reviewer #2

(Remarks to the Author)

In this very interesting study, de Queiroz and co-authors have conducted synaptosomal RNA profiling followed by a series of validation experiments to identify a population of mRNAs that are localized to distal axons and presynaptic compartments of *Drosophila* Mushroom Body Memory Neurons. They further investigate the role of the RNA binding protein Imp1 and its prion-like domain in this process of axonal RNA localization which is also necessary for long-term memory. The authors use CLIP-Seq to profile and identify Imp1 target mRNAs and show the role of 3'UTR sequences using reporters to promote axonal mRNA localization for a few candidates. A major strength of the study is the new and detailed characterization of synaptic mRNAs in this *Drosophila* model system and other experiments suggesting a potential mechanism that links axonal RNA localization to long term memory. A weakness of the paper is that data shown fall short of making these mechanistic connections for a RNA binding protein to axonal RNA localization and long term memory. Further experiments which are doable seem necessary as outlined below to provide more compelling support for the model and the conceptual advance.

The authors provide two novel data set that support the broad role of Imp in axonal RNA localization: 1) synaptosome RNA profiling identified hundreds of enriched mRNAs, 2) iCLIP identification of Imp mRNA targets, and overlap of the data sets, followed by validation. The authors show that three Imp targets are mislocalized in axons in the Imp PLD deletion mutant line (Fig. 6C-E). It seems that the missing data set to provide a more unequivocal conclusion would be to perform synaptosomal RNA profiling on the Imp PLD mutant. This experiment and analysis should be performed. The expectation is that Imp targets are not enriched at synapses.

More FISH data is needed to analyze mRNA mislocalization in the Imp PLD mutant. The validation of mRNA localization impairments for the three mRNA targets shown in Fig. 6C-E is supportive, but needs to be expanded and followed up, as this a major part of the proposed model and data interpretation. Can an experiment be done to validate a few more mRNA targets, and perhaps show impairments in distal enrichment along axons, for example, using the paradigm shown in Fig. 5 applied to the Imp PLD mutant?

A previous paper from this group (Vijayakumar et al Nat. Comm. 2019) has shown that the Imp PLD has two independent functions in regulating dynamics of Imp RNP granule assembly and axonal transport. In that study, the role of Imp in granule dynamics and transport was uncoupled through the analysis of other Imp PLD mutants (N-terminal fusion of PLD). As the present study seeks to make a strong link of Imp to axonal RNA transport, the authors may need to at least discuss the potential role of this second function on RNA granule assembly.

Some evidence is needed on reduced axonal levels of the encoded proteins in the Imp PLD mutant axons and/or synaptic fractions. This is necessary because the authors seek to make a connection between axonal mRNA localization and protein synthesis dependent long-term memory.

The authors should analyze for Imp expression in the synaptosome fraction, both wild type and Imp PLD mutant, but western blot.

Is Imp present in the distal axonal fields shown in Fig. 5?

qRT-PCR can be done to provide molecular validation that a few Imp target mRNAs are reduced in synaptosomes from Imp PLD mutants compared to controls.

(Remarks on code availability)

Reviewer #3

(Remarks to the Author)

The necessity and operational mechanisms of synaptic mRNA localization in memory formation in vivo have long been enigmatic. Such selective recruitment hinges on distinct 3' UTR sequences and trans-acting RBPs, proposed to serve as markers for activated synapses critical in memory formation. However, whether this localized synaptic mRNA recruitment occurs within native memory circuits and in response to LTM-inducing stimuli remains predominantly obscure.

Utilizing a multidisciplinary approach, the authors explored axonal mRNA localization in *Drosophila* Mushroom Body γ neurons. RNA profiling revealed an axonal enrichment of numerous mRNAs. Single molecule FISH experiments unveiled diverse mRNA localization patterns in axonal terminals, with selective accumulation in a specific MB subcompartment. Functional analyses identified a role of the RNA binding protein Imp in mediating the transport of a subset of mRNAs to axons. They find that an *imp- Δ PLD* mutant which disrupts the axonal mRNA localization of a subset of messages, they demonstrate a necessity for long-term courtship memory, in result suggesting a critical role of local axonal RNA regulation in memory consolidation.

The functional significance of local translation in behavioral control has posed significant challenges due to the absence of components exclusively dedicated to local axonal translation. The question of whether local translation of synaptically

targeted mRNAs is essential for establishing long-term, protein synthesis-dependent memories has endured. Therefore, this study is inherently significant in addressing this longstanding question, and it presents an interesting finding with the use of the *imp-ΔPLD* mutant.

Major points:

They observe selective accumulation of certain mRNAs in a distal axonal sub-compartment innervated by specific input modulatory neurons, potentially indicating local accumulation in response to circuit activity. However, it is plausible that this phenomenon could also stem from molecular diversity across synapse populations. Nonetheless, it should be feasible to investigate whether this sub-compartment-specific enrichment is influenced by long-term memory (LTM)-mediating conditioning. Additionally, testing for the involvement of dopamine neurons (DANs) using activity tools could complement their findings and enhance the impact of their manuscript.

Related: concerning Fig. 5 and the gamma-compartment enrichment: do I understand correctly that they are using a probe set against *arc1* in a generic manner (instead of driving a probe in a gamma-KC specific manner)? If so, how can they be sure that their smFISH signals do indeed derive from KCs and not from MBONs or DANs?

In their initial results, there is a clear emphasis on synaptic and mitochondrial messages. However, their mechanistic exploration of IMP's role primarily centers on specific mRNAs (*profilin*, *actin5C*, *arc1*), none of which seem to be authentic synaptic or mitochondrial messages. Addressing this discrepancy could be achieved through a table summarizing IMP-dependence for the transcripts they've examined by FISH. This would provide clarity and aid in reconciling their findings.

The IMP iCLIP data are per se impressive concerning their 3-UTR specificity. Still, the overlap between IMP iCLIP bound and IMP RIP is rather moderate. Can they offer an explanation in this point?

Understanding the findings from their previous IMP-related papers and distinguishing them from the results of this new analysis presented a challenge. It would be beneficial for them to review their manuscript with this consideration in mind, ensuring clarity and coherence in delineating the outcomes of the current study in relation to their previous research on IMP. Incorporating a succinct paragraph addressing this point could enhance the clarity and comprehensibility of their manuscript.

They say that they used an optimized protocol based on differential centrifugation and discontinuous sucrose gradient to isolate synaptosome fractions. If they optimized, then there should be citable references which they might like to incorporate.

(Remarks on code availability)

Version 1:

Reviewer comments:

Reviewer #1

(Remarks to the Author)

In this carefully revised manuscript by Besse and coworkers, the authors made a genuine and impressive effort to address the points by all three referees. Overall, although not all experiments worked out for them (not to be expected in the first place), this led to the addition of a series of new display items, both in the main manuscript, in the Supplements and impressively of 4 figures for the referees. Let me select a couple of those new experiments. First, the revised Fig. 5D provides convincing evidence that silencing local circuit activity inhibits local enrichment of *arc* mRNA suggesting that this might induce/establish subdomains along the axons. This is clearly important and a major addition to the manuscript. Second, I agree with the authors and their new findings in Figure I that it does not come as surprise that no major changes in protein levels were found. Of course, I also agree with them that hopefully the SunTag might bring the expected results. A second and important aspect is the extension of the list of synaptic RNAs in their system and the newly added mRNAs, e.g. *eIF4A*, *cg15098* and *eIF4E1/bnb* and *crc*. This is clearly additional support for their main findings. Third, I would like to highlight that in the revised version, the fact that *Imp-deltaPLD* protein is not efficiently transported to axons is important (revised Fig. S3B. I would easily see that this might be even be shown in the main figures. Finally, I like their preliminary findings on *arc* mRNA presented in Fig. III. This is promising and hope that the observed trend might eventually hold in the future.

Together, this study clearly improved during revision. I am glad that some of the suggested experiments worked out for the authors. Consequently, I suggest publication of this important and timely study, which is a clear advance for the field.

(Remarks on code availability)

Reviewer #2

(Remarks to the Author)

The authors have been very responsive to suggestions for new experiments and analysis. In particular, new data are provided to validate additional targets identified by transcriptomic analysis. The authors explanations and clarifications are acceptable. This is a very nice paper that will advance the field.

(Remarks on code availability)

Reviewer #3

(Remarks to the Author)

They now went through an extended revision experimentally addressing the suggestions of the other reviewers and mine. At least in one case (role of compartment-specific dopaminergic innervation), this allowed for the addition of new interesting data. I now support the publication of the manuscript unreservedly.

(Remarks on code availability)

ANSWER to the REVIEWER'S COMMENTS

We thank our referees for their thorough analysis of our article, and for their comments and questions, which helped us significantly improve our manuscript.

In the revised version, we have extended our analysis of the Imp regulon, validating and quantitatively characterizing *in vivo* the behavior of 10 different transcripts. Our new results further strengthened the conclusion that the axonal localization of selected transcripts is altered in MB γ neurons upon inhibition of Imp transport (*imp- Δ PLD*).

Additionally, we have significantly strengthened our analysis of the novel compartment-specific mRNA localization we had uncovered. First, we identified additional mRNAs that exhibit a distal axonal enrichment and analyzed the dependence of this enrichment on 3'UTR sequences. Second, our new results demonstrate that compartment RNA localization depends on the activity of upstream modulatory neurons. These latter results now provide strong *in vivo* evidence for the importance of local circuit activity in establishing transcriptomically distinct sub-domains along the axons of memory neurons.

These new results, together with others described below, and with the re-organization of the main text we have performed, have resulted in an improved version, where we address most of the points raised by the referees. Three supplementary Figures, one supplementary Table and several Figure panels were added, resulting in a manuscript containing 7 main Figures and 10 supplementary ones.

Reviewer #1 -----

In the interesting and timely manuscript entitled Axonal RNA localization is essential for long-term memory, de Queiroz et al. identify mRNA targets in synaptosomes of *Drosophila* and confirm selected transcript localization in MB axons. Moreover, the authors identify a subset of mRNAs localized by their 3'-UTR via the Imp RNA-binding protein. Disruption of Imp transport resulted in mRNA mislocalization and deficiencies in LTM during male *Drosophila* courtship conditioning.

Overall, the data is presented understandably and in a conclusive manner. The experiments are explained well and interpreted rigorously. **The results expand nicely on previous research of the Besse lab and add valuable new insight to the community.** Enclosed are suggestions that might expand the impact of this interesting and original study.

Main comments

- Is the compartment-specific γ 5 mRNA accumulation (Fig. 5) also true for *act5c* mRNA?

We initially did not plot the γ 5 accumulation of *act5C* mRNA as comparing distal vs proximal densities gets problematic when the total number of axonal smFISH spots is very low. To still address the referee's comment, we filtered the *act5C* samples and eliminated those with a number of axonal spots below a fixed threshold (see Materials and Methods). Quantifying compartment-specific γ 5 mRNA accumulation in this condition revealed that *act5C* shows a distal accumulation bias (see revised Figure 5C, where we have also extended our analysis of γ 5 enrichment to all Imp-bound mRNAs analyzed).

While profilin mRNA accumulates in this compartment, this is not clearly seen in Fig. 4D. Could it be that the 3'-UTR is sufficient for axonal, but not compartment-specific localization?

We thank the referee for his/her comment and question. The measure we defined to describe the differential accumulation of transcripts in the γ 5 compartments corresponds to a ratio of densities (number of smFISH spots per μm^3) and thus does not reflect absolute abundance. In the case of *profilin*, the overall abundance in axons is low, which explains why the signal seen in the γ 5 compartment in Figure 4D was not high. The density of spots visible in this compartment is however higher than that seen in the more proximal compartments (see quantification in Figure 5C). To address the referee's comment, we modified Figure 5 legend to clarify our definition of enrichment and also replaced the image shown in Figure 4D with an image of the *pabp* reporter (higher distal signal).

Whether compartment-specific localization is 3'UTR-dependent is an interesting question that we addressed by generating 3 new *gfp*-3'UTR transgenic lines (UAS-*gfp-pabp*-3'UTR, UAS-*gfp-lk6*-3'UTR, UAS-*gfp-arc1*-3'UTR). As shown in our revised Figure S5B, quantification of *gfp* smFISH signal densities in distal vs proximal compartments revealed a relative enrichment in the γ 5 compartment compared to more proximal compartments for all constructs. The γ 5 enrichment observed for *gfp*-3'UTR constructs is however lower than that observed for endogenous mRNAs, suggesting that other factors (5'UTRs,

RNA processing etc) might contribute to enrichment. These data are presented and discussed in the revised manuscript.

- The finding that localized mRNAs may play a role in local circuits (Fig. 5) is fascinating, but not extensively explored in this study. Though a difficult experiment, it would be interesting to see whether the local circuitry indeed is responsible for axonal mRNA localization.

Would it be possible to interfere/inhibit this circuit in brains?

To test whether the local enrichment of *arc1* mRNA is induced by the activity of the PAM- γ 5 DAN neurons that selectively innervate the γ 5 compartment of MB γ axons, we silenced these neurons through Kir2.1-mediated hyperpolarization. Specifically, we used the MB315C-Gal4 to induce selective expression of the Kir2.1 channel^{1,2} and analyzed in this context the localization of *arc1* mRNA via smFISH. As shown in our revised Figure 5D, silencing the upstream PAM- γ 5 DANs indeed inhibited the local enrichment of *arc1* mRNA, suggesting that the compartment-specific recruitment of *arc1* mRNA is induced in response to the activity of local circuits known for their involvement in LTM consolidation. This new result provides strong *in vivo* evidence for the importance of local circuit activity in establishing transcriptomically distinct sub-domains along the axons of memory neurons.

Or would the circuit functionality be affected by mRNA mislocalization (e.g. by mutated Imp)?

As shown by the previous work of the Keleman group³, the same recurrent circuit (PAM- γ 5 Dopaminergic neurons \rightarrow MB γ neurons \rightarrow MBON- γ 5 β '2a output neuron) is used for both short-term (STM) and long-term (LTM) courtship memory. Defective STM would thus be expected if the circuit was affected in the *imp- Δ PLD* mutant, which is not the case.

This result, together with our quantification of presynaptic active zone number and enrichment in the presynaptic structural component Brp (see our answer to a following point below), thus indicate that the circuit is not functionally affected in the *imp- Δ PLD* mutant. Whether LTM-induced structural plasticity is altered in the *imp- Δ PLD* mutant is an open question that will be interesting to address in the future.

Alternatively, would the protein product of localized mRNA be relevant for the circuit and is it specifically localized in γ 5 as well?

To address the referee's comment, we tried to measure protein amounts through antibody staining performed on whole adult brain (see Figure I) for genes in which antibodies or knock-in lines were available (ie *arc1*, *profilin* and *pabp*). While significant signal was observed in the γ 5 compartment (Figures I and II), no differences could be observed when comparing the average intensities in the γ 5 and γ 3-4 compartments in this condition. This does not come as a surprise as we expect proteins to be produced specifically, and perhaps transiently, in response to LTM stimuli.

Figure I. Relative intensities of Arc1, Profilin and PABP protein signals in the γ 5 and γ 3-4 compartments.

Arc 1 and Profilin signal intensities were measured from confocal images of whole mount brains immunostained with antibodies raised against endogenous proteins.

PABP signal intensity was measured from confocal images of *pabp*-GFP whole mount brains immunostained with anti-GFP antibodies.

Signals were normalized for each brain to the value of the γ 5 compartment.

- As the total mRNA levels are not altered in PLD mutants, but axonal localization is disrupted for some targets (Fig. 6C-E), would mRNA accumulate in the cell body or other compartments instead? Could this be seen in FISH images?

It is to be noticed that the proportion of mRNA molecules found in the axonal lobes is very small compared to the proportion of molecules found in cell bodies (estimated as less than 1% for most

mRNAs, similar to what is described in vertebrates). Axonal mRNA mislocalization is thus not expected to have a measurable impact on the cell body content.

• It would be interesting to investigate how RNA mislocalization affects courtship behavior. Is translation affected as well? Is the protein product mislocalized?

Our model is that the axonal translation of Imp-bound mRNAs is triggered specifically, and perhaps transiently, in response to LTM-inducing stimuli and we thus do not expect a noticeable impact on steady-state levels of proteins. Still, to address the referee's point, we performed immunostainings on control and *imp-ΔPLD* whole mount brains to detect Arc1 and Profilin and measured signal intensities in MB γ axons from confocal images. As shown in Figure II, no significant differences were observed when comparing control and *imp-ΔPLD* conditions, which is consistent with the idea that local translation of the Imp-bound mRNAs under study is induced in specific contexts.

Figure II. Relative intensities of Arc1 and Profilin in control (ctrl) and *imp-ΔPLD* (Δ PLD) conditions.

Arc1 and Profilin signal intensities were measured from confocal images of whole mount brains immunostained with antibodies raised against endogenous proteins.

Signal were measured on the γ 5 compartment, but similar results were obtained for the γ 3- γ 4 compartments.

Three replicate experiments were performed and signals were normalized for each experiment to 100 (average value of the control condition).

Carefully testing whether the activity-dependent translation of localized mRNAs is affected in the *imp-ΔPLD* context would require more elaborate readouts (e.g. Suntag reporters) and optimized LTM-mimicking stimuli (e.g. optogenetic activation of PAM- γ 5-DANs in flies conditioned for STM), which are yet to be developed.

Is the local circuit or are synapses altered (e.g. fewer functional synapses by immunostaining)?

As mentioned in our answer to a previous point, Short-Term Memory is not impacted by the axonal mRNA mislocalization induced by the *imp-ΔPLD* mutation, indicating no major alteration of the local circuit. To confirm this and quantify the number of presynaptic active zones detected in control and *imp-ΔPLD* individuals, we performed immunostaining against the Brp protein, an essential structural component of presynaptic active zones whose incorporation at presynapse was shown to be modulated in various synaptic plasticity contexts⁴. No significant differences in the number of detected active zones or in the amount of recruited BRP molecules was observed between control and *imp-ΔPLD* mutants (revised Figures S9A,B). These results suggest that the *imp-ΔPLD* mutation does not majorly alter circuit and/or synapse organization.

Minor comments

• Please state for all graphs what the error bars represent in the figure legends (e.g. Fig. 2F, Fig. 7).
We have added the missing information. Error bars represent s.e.m.

• How does the axonal localization of mRNAs presented in Fig. 3B compare to their abundance in the cell body (as done in Fig. 2F)?

We have generated a correlation plot showing abundance in axons in function of abundance in cell body for all Imp targets analyzed (displayed in revised Figure S3C).

For easier comparison, could the authors add the means to Fig. 3B.

Means are now shown in our revised Figure 3B.

Out of interest, how strongly were the 13 overlapping targets enriched in synaptosomes?

We have now included a Table summarizing synaptosomal enrichment and smFISH results for all analyzed Imp targets (revised Table S3).

• In some images, it appears that mRNA is quite abundant in areas adjacent to the axonal compartment (e.g. Fig. 3). Though MB γ axons are clearly the focus of this study, could the authors comment on the surrounding mRNA localization, if relevant? Are these adjacent areas anatomically connected/related to the axonal compartment?

The large bundle formed by the hundreds of MB γ axons is surrounded by the cell bodies of other neuronal populations (signal typically found in the “corners” of the images and often abundant), by the axons of other MB populations ($\alpha\beta$ and $\alpha'\beta'$), by the cellular processes of non-MB neuronal populations, some anatomically connected to MB axons (e.g. Dopaminergic or Octopaminergic DANs, MB output neurons), as well as by the cellular processes of glial cells. While the precise origin of the signal from neighboring cellular processes is very difficult to identify without complementary markers, signals in neighboring cell bodies is easy to recognize. We have thus marked with asterisks such signal in images where this may have brought confusion (Figures 2,3,5; Figures S2).

By the way, the scale bar is missing in Fig. 3.

Thanks for noticing, this has been corrected.

• For the data in Fig. 4E, could the authors provide example images for all conditions (main or Suppl figure)?

We are now showing in revised Figure S5A example images of the *gfp* smFISH signal produced the different *gfp*-3'UTR transgenes.

• As Imp iClip sites are enriched in 3'-UTRs (Fig. 4), is this true for all 13 overlapping synaptosomal targets presented in Fig. 3A, or is there variation?

In the course of revision work, we optimized our iCLIP analysis pipeline (removal of duplication), which generated a globally similar, but more accurate, list of Imp-bound mRNAs (see revised Table S5). This optimization improved the number of “overlapping synaptosomal targets” represented in Figure 3A (24 mRNAs instead of 13 initially), while preserving the very strong enrichment in 3'UTR binding. Preferential 3'UTR binding is true for all annotated overlapping mRNAs, and this is now shown in the revised Figure S4.

Reviewer #2 -----

In this very interesting study, de Queiroz and co-authors have conducted synaptosomal RNA profiling followed by a series of validation experiments to identify a population of mRNAs that are localized to distal axons and presynaptic compartments of *Drosophila* Mushroom Body Memory Neurons. They further investigate the role of the RNA binding protein Imp1 and its prion-like domain in this process of axonal RNA localization which is also necessary for long-term memory. The authors use CLIP-Seq to profile and identify Imp1 target mRNAs and show the role of 3'UTR sequences using reporters to promote axonal mRNA localization for a few candidates. **A major strength of the study is the new and detailed characterization of synaptic mRNAs in this *Drosophila* model system and other experiments suggesting a potential mechanism that links axonal RNA localization to long term memory.** A weakness of the paper is that data shown fall short of making these mechanistic connections for a RNA binding protein to axonal RNA localization and long term memory. Further experiments which are doable seem necessary as outlined below to provide more compelling support for the model and the conceptual advance.

The authors provide two novel data set that support the broad role of Imp in axonal RNA localization: 1) synaptosome RNA profiling identified hundreds of enriched mRNAs, 2) iCLIP identification of Imp mRNA targets, and overlap of the data sets, followed by validation. The authors show that three Imp targets are mislocalized in axons in the Imp PLD deletion mutant line (Fig. 6C-E). It seems that the missing data set to provide a more unequivocal conclusion would be to perform synaptosomal RNA profiling on the Imp PLD mutant. This experiment and analysis should be performed. The expectation is that Imp targets are not enriched at synapses.

We thank the referee for his thoughtful comments and suggestions. We would like to point out however that the synaptosome analysis shown in Figure 1 was performed from lysates of entire fly heads and thus represents a bulk analysis, in which the content of all synapses of the adult fly brain was analyzed by RNA-seq. Given that the Imp protein is targeted specifically to the axons of MB γ neurons (⁵ and revised Figure S3A), and that the number of these neurons is small compared to the entire number of adult brain neurons (1,350 MB γ neurons for about 130,000 neurons total, *i.e.* about 1% of the total), RNA molecules depending on Imp for their localization to MB γ axons are not expected to be identified through comparison of wild-type *vs imp- Δ PLD* bulk synaptosomes. In the revised version of the manuscript, we have better highlighted that Imp is selectively transported to the axons of MB γ neurons (see also revised Figure S3A).

More FISH data is needed to analyze mRNA mislocalization in the Imp PLD mutant. The validation of mRNA localization impairments for the three mRNA targets shown in Fig. 6C-E is supportive, but needs to be expanded and followed up, as this a major part of the proposed model and data interpretation. Can an experiment be done to validate a few more mRNA targets, and perhaps show impairments in distal enrichment along axons, for example, using the paradigm shown in Fig. 5 applied to the Imp PLD mutant?

To address the referee's comment, we aimed at validating the behavior of few more Imp-bound mRNAs. This process involved i- selection of candidates based on their presence in synaptosomes, overall abundance and binding to Imp; ii- refinement of candidate's list based on the possibility to design robust smFISH probe sets (a fraction of the candidates are eliminated based on transcript length, CG content or redundant sequences); iii- test of the probe sets on whole mount brains and estimation of smFISH spot numbers.

Out of the 12 genes we initially selected to address the referee's point, 5 met the criteria listed above (*eIF4A*, *cg15098*, *eIF4E-1*, *bnb*, *crc*). For those 5 genes, we then quantitatively analyzed and compared axonal smFISH signals in control and *imp- Δ PLD* mutants. As shown in our revised Figure 6, *eIF4A* and *cg15098* mRNAs showed *imp*-dependent localization to axons while *eIF4E-1*, *bnb*, *crc* mRNAs showed *imp*-independent localization to axons. Together, this work both confirmed and strengthened our previous conclusions: in total, we could validate the axonal localization of 10 Imp-bound RNAs and showed that axonal localization of half of them is dependent on Imp, raising interesting new questions related to the additional factors (RBPs, RNA modifications ...) acting together with Imp to promote axonal RNA localization.

A previous paper from this group (Vijayakumar et al Nat. Comm. 2019) has shown that the Imp PLD has two independent functions in regulating dynamics of Imp RNP granule assembly and axonal transport. In that study, the role of Imp in granule dynamics and transport was uncoupled through the analysis of other Imp PLD mutants (N-terminal fusion of PLD). As the present study seeks to make a strong link of Imp to axonal RNA transport, the authors may need to at least discuss the potential role of this second function on RNA granule assembly.

This is a good idea, we are now discussing this aspect in link to the differential sensitivity of Imp-bound mRNAs to the *imp- Δ PLD* mutation.

Some evidence is needed on reduced axonal levels of the encoded proteins in the Imp PLD mutant axons and/or synaptic fractions. This is necessary because the authors seek to make a connection between axonal mRNA localization and protein synthesis dependent long-term memory.

That localized mRNAs may serve as a substrate for local protein synthesis is indeed a model we discuss in our discussion section. We however think that the axonal translation of Imp-bound mRNAs is triggered specifically, and perhaps transiently, in response to LTM-inducing stimuli, and thus do not expect differences in steady-state protein amounts in the *imp- Δ PLD* mutant. Consistent with this idea, no significant differences were observed between control and *imp- Δ PLD* mutants when comparing signal intensities obtained after immunostaining of whole mount brains for proteins encoded by mRNAs localizing in an *imp- Δ PLD*-dependent manner (*Arc1* and *Profilin*; see Figure II and the corresponding answer to Referee#1).

Quantitatively assessing local protein synthesis in a whole brain context requires more elaborated and quantitative methods that are yet to be implemented (super-resolution for synapse-level analysis, axonally-localized SunTag reporters, sub-cellular live-imaging in response to optogenetic activation etc...).

The authors should analyze for Imp expression in the synaptosome fraction, both wild type and Imp PLD mutant, but western blot.

To address this point, we collected synaptosomal fractions from GFP-Imp and GFP-Imp- Δ PLD knock-in lines and performed Western-Blot experiments on the recovered fractions. These experiments first demonstrated that wild-type Imp can indeed be detected in synaptosome fractions. They also revealed that the Imp- Δ PLD protein is poorly detected at synapses, consistent with our previous observation that the Imp- Δ PLD protein is not efficiently transported to axon terminals in MB γ neurons⁶. We have included these new data in Supplementary Figure S3B and refer to them in the main result section.

Is Imp present in the distal axonal fields shown in Fig. 5?

Yes, Imp is present all along MB γ axons, including in the γ 5 compartment. To address the referee's question, we have included an image showing the presence of endogenous GFP-Imp in MB γ axons, where we highlighted the γ 5 compartment (see revised Figure S3A).

qRT-PCR can be done to provide molecular validation that a few Imp target mRNAs are reduced in synaptosomes from Imp PLD mutants compared to controls.

We thank the referee for his/her suggestion. As explained in our answer to the first point, our synaptosome fractions are however obtained from entire fly heads and thus represent bulk experiments in which a variety of neuron cell types are mixed. Dissecting the contribution of Imp requires spatial resolution and this is the reason why we focused on quantitative analyses of smFISH signals on 3D labeled brains.

Reviewer #3

The necessity and operational mechanisms of synaptic mRNA localization in memory formation in vivo have long been enigmatic. Such selective recruitment hinges on distinct 3' UTR sequences and transacting RBPs, proposed to serve as markers for activated synapses critical in memory formation. However, whether this localized synaptic mRNA recruitment occurs within native memory circuits and in response to LTM-inducing stimuli remains predominantly obscure.

Utilizing a multidisciplinary approach, the authors explored axonal mRNA localization in *Drosophila* Mushroom Body γ neurons. RNA profiling revealed an axonal enrichment of numerous mRNAs. Single molecule FISH experiments unveiled diverse mRNA localization patterns in axonal terminals, with selective accumulation in a specific MB subcompartment. Functional analyses identified a role of the RNA binding protein Imp in mediating the transport of a subset of mRNAs to axons. They find that an imp- Δ PLD mutant which disrupts the axonal mRNA localization of a subset of messages, they demonstrate a necessity for long-term courtship memory, in result suggesting a critical role of local axonal RNA regulation in memory consolidation.

The functional significance of local translation in behavioral control has posed significant challenges due to the absence of components exclusively dedicated to local axonal translation. The question of whether local translation of synaptically targeted mRNAs is essential for establishing long-term, protein synthesis-dependent memories has endured. **Therefore, this study is inherently significant in addressing this longstanding question, and it presents an interesting finding with the use of the imp- Δ PLD mutant.**

Major points:

They observe selective accumulation of certain mRNAs in a distal axonal sub-compartment innervated by specific input modulatory neurons, potentially indicating local accumulation in response to circuit activity. However, it is plausible that this phenomenon could also stem from molecular diversity across synapse populations. Nonetheless, it should be feasible to investigate whether this sub-compartment-specific enrichment is influenced by long-term memory (LTM)-mediating conditioning.

We thank the referee for his/her comment and suggestion. To address this point, we aimed at comparing the recruitment of *arc1* mRNA to the γ 5 compartment in naïve flies and flies that experienced interspaced cycles of courtship conditioning to generate LTM. We used courtship conditioning because it was shown

to depend both on MB γ neurons and on the specific activation of Pam- γ 5 DANs, which innervate the distal γ 5 compartment^{3,7}. *arc1* mRNA distribution was analyzed at two time points after training (30 minutes and 2 hours).

As shown in Figure III, a tendency for increased accumulation of *arc1* mRNA in γ 5 compartment was observed in response to conditioning, especially 30 minutes after conditioning. This result, however, was not fully reproducible and statistical significance could not be reached. This might be due to the fact that our naïve flies actually exhibited a significant distal enrichment to start with (perhaps because of the spontaneous activity of the circuit). Given the absence of significant difference, we did not integrate these data into the manuscript. The new results we have obtained when playing with the activity of upstream γ 5 DANs (discussed in our answer to the next point) however clearly point to the importance of upstream dopaminergic neuron stimulation in the recruitment of RNAs to the γ 5 compartment.

Figure III. Accumulation of *arc1* mRNA into the distal γ 5 compartment in response to courtship conditioning.

Distal enrichment is quantified as the ratio of the densities of *arc1* mRNA spots in γ 5 vs more proximal compartments.

smFISH experiments were performed 30 min (left) or 2 hours (right) after interspaced training cycles.

Three to four replicates were performed and data points color-coded based on the replicate they belong to.

Statistical tests: Mann-Whitney tests.

Additionally, testing for the involvement of dopamine neurons (DANs) using activity tools could complement their findings and enhance the impact of their manuscript.

We agree with the referee and have performed experiments to silence the upstream dopaminergic neurons innervating the γ 5 compartment. As a similar question was raised by referee #1, we copy below our answer to referee #1 question:

*To test whether the local enrichment of arc1 mRNA is induced by the activity of the PAM- γ 5 DAN neurons that selectively innervate the γ 5 compartment of MB axons, we silenced these neurons through Kir2.1-mediated hyperpolarization. Specifically, we used MB315C-Gal4 to induce selective expression of the Kir2.1 channel^{1,2} and analyzed in this context the localization of *arc1* mRNA via smFISH. As shown in our revised Figure 5D, silencing the upstream PAM- γ 5 DANs indeed inhibited the local enrichment of *arc1* mRNA, suggesting that the compartment-specific recruitment of *arc1* mRNA is induced in response to the activity of local circuits known for their involvement in LTM consolidation. This new result provides strong in vivo evidence for the importance of local circuit activity in establishing transcriptomically distinct sub-domains along the axons of memory neurons.*

Related:

concerning Fig. 5 and the gamma-compartment enrichment: do I understand correctly that they are using a probe set against *arc1* in a generic manner (instead of driving a probe in a gamma-KC specific manner)? If so, how can they be sure that their smFISH signals do indeed derive from KCs and not from MBONs or DANs?

We indeed used smFISH probes recognizing the endogenous *arc1* mRNA sequence in our experiments. As described in our material and methods section, however, careful 3D reconstructions of the MB γ lobe were performed in Imaris using the GFP signal driven by a MB γ -specific Gal4, and smFISH spots sitting outside the lobe surface were excluded from analysis.

As MBONs and DANs dendritic and axonal branches can intermingle with MB axonal branches, we aimed at further validating the origin of *arc1* mRNA signal using two different approaches. First, we aimed at degrading *arc1* mRNA specifically in MB γ neurons by combining UAS-RNAi constructs with the 71G10-Gal4 driver (stronger and more specific than 201Y-Gal4). As shown in Figure IV, efficient

RNAi-mediated degradation could however not be reached with any of the UAS-RNAi line we used (TRIP line BL#25954, VDRC lines #48131 & 25954).

Figure IV. Quantification of *arc1* mRNA signal in control (ctrl) and *arc*-RNAi conditions.

Three different UAS-RNAi lines were tested in combination with the MB γ specific-Gal4 GMR71G10, and their efficiency assessed by anti-Arc1 immunostaining on adult brain.

No difference was observed when comparing cells expressing the RNAi control to control ones (Mann-Whitney tests).

As an alternative, we generated transgenic lines in which an exogenous *gfp-arc1* 3'UTR construct was expressed in MB γ neurons using the UAS/Gal4 system. Analyzing the distribution of *gfp-arc1* 3'UTR RNAs expressed specifically in MB neurons first revealed the presence of *gfp* smFISH spots in axons (Figure S5A), supporting the idea that *arc1* 3'UTR drive the axonal localization of RNAs expressed in MB γ neurons. Quantification of distal: proximal *gfp* smFISH signal in this context further showed γ 5 enrichment (Figure S5B), supporting the idea that *arc1* mRNA expressed from MB neurons has an intrinsic capacity for local enrichment. γ 5 enrichment was however less prominent for *gfp-arc1* 3'UTR than for endogenous *arc1*, suggesting the existence of additional factors. This is discussed in the revised version of the manuscript.

In their initial results, there is a clear emphasis on synaptic and mitochondrial messages. However, their mechanistic exploration of IMP's role primarily centers on specific mRNAs (profilin, actin5C, *arc1*), none of which seem to be authentic synaptic or mitochondrial messages. Addressing this discrepancy could be achieved through a table summarizing IMP-dependence for the transcripts they've examined by FISH. This would provide clarity and aid in reconciling their findings.

mRNAs to be visualized by smFISH were selected based on two independent datasets: synaptosome RNA-seq and Imp iCLIP. While mRNAs selected in Figure 2 were aimed at representing the different functional categories identified in our synaptosome RNA-seq, mRNAs displayed in Figure 3 were selected based on their presence in synaptosome RNA-seq and Imp interactomes. It is to be noted that the mitochondrial mRNAs identified in our synaptosome RNA-seq were all too short, preventing us for designing an optimal number of primary probes (>28) for smFISH analyses.

As suggested by the referee, and to improve the clarity of our manuscript, we have generated an additional Table (revised Table S3) that summarizes the characteristics of mRNAs mentioned in our study.

The IMP iCLIP data are per se impressive concerning their 3-UTR specificity. Still, the overlap between IMP iCLIP bound and IMP RIP is rather moderate. Can they offer an explanation in this point?

Although RIP and iCLIP experiments both rely on initial immunoprecipitation of the RNA Binding protein of interest, they each have their technical specificities (e.g. loss of weak affinity binding for RIP, nucleotide bias for crosslinking in CLIP, suboptimal efficiency for both ⁸). A partial overlap is thus classically observed when comparing such kind of datasets. In our case, it is to be further noted that the iCLIP experiments we performed were based on immunoprecipitation of GFP-tagged Imp proteins expressed in the entire brain (GFP-Imp protein trap line), while the RIP experiments were performed through immunoprecipitation of Flag-tagged Imp expressed more specifically in MB γ neurons (201Y-Gal4>Flag-Imp). mRNAs specifically expressed and/or bound in MB γ neurons are thus expected to generate low/no signal in iCLIP. Conversely, RNAs bound in all cell types but with weak affinity may be

lost in the RIP-seq procedure and observed specifically in the iCLIP dataset. Last, two different technologies were used for recovery of RNA identity (Genechip for the RIP and RNA-seq for the iCLIP).

Understanding the findings from their previous IMP-related papers and distinguishing them from the results of this new analysis presented a challenge. It would be beneficial for them to review their manuscript with this consideration in mind, ensuring clarity and coherence in delineating the outcomes of the current study in relation to their previous research on IMP. Incorporating a succinct paragraph addressing this point could enhance the clarity and comprehensibility of their manuscript.

We carefully went through the manuscript and tried to better highlight what was shown in our previous study ⁶ and what are the new outcomes of this manuscript.

They say that they used an optimized protocol based on differential centrifugation and discontinuous sucrose gradient to isolate synaptosome fractions. If they optimized, then there should be citable references which they might like to incorporate.

We totally agree with the referee and have added the citation in the main result section (the reference was initially only mentioned in the Materials and Method section).

References

1. Baines, R.A., Uhler, J.P., Thompson, A., Sweeney, S.T., and Bate, M. (2001). Altered electrical properties in *Drosophila* neurons developing without synaptic transmission. *J Neurosci* 21, 1523-1531. 10.1523/JNEUROSCI.21-05-01523.2001.
2. Otto, N., Pleijzier, M.W., Morgan, I.C., Edmondson-Stait, A.J., Heinz, K.J., Stark, I., Dempsey, G., Ito, M., Kapoor, I., Hsu, J., et al. (2020). Input Connectivity Reveals Additional Heterogeneity of Dopaminergic Reinforcement in *Drosophila*. *Curr Biol* 30, 3200-3211 e3208. 10.1016/j.cub.2020.05.077.
3. Kruttner, S., Traunmuller, L., Dag, U., Jandrasits, K., Stepien, B., Iyer, N., Fradkin, L.G., Noordermeer, J.N., Mensh, B.D., and Keleman, K. (2015). Synaptic Orb2A Bridges Memory Acquisition and Late Memory Consolidation in *Drosophila*. *Cell Rep* 11, 1953-1965. 10.1016/j.celrep.2015.05.037.
4. Van Vactor, D., and Sigrist, S.J. (2017). Presynaptic morphogenesis, active zone organization and structural plasticity in *Drosophila*. *Curr Opin Neurobiol* 43, 119-129. 10.1016/j.conb.2017.03.003.
5. Medioni, C., Ramialison, M., Ephrussi, A., and Besse, F. (2014). Imp promotes axonal remodeling by regulating profilin mRNA during brain development. *Curr Biol* 24, 793-800. 10.1016/j.cub.2014.02.038.
6. Vijayakumar, J., Perrois, C., Heim, M., Bousset, L., Alberti, S., and Besse, F. (2019). The prion-like domain of *Drosophila* Imp promotes axonal transport of RNP granules in vivo. *Nat Commun* 10, 2593. 10.1038/s41467-019-10554-w.
7. Keleman, K., Vrontou, E., Kruttner, S., Yu, J.Y., Kurtovic-Kozaric, A., and Dickson, B.J. (2012). Dopamine neurons modulate pheromone responses in *Drosophila* courtship learning. *Nature* 489, 145-149. 10.1038/nature11345.
8. Hafner, M., Katsantoni, M., Köster, T., Marks, J., Mukherjee, J., Staiger, D., Ule, J., and Zavolan, M. (2021). CLIP and complementary methods. *Nat Rev Methods Primers* 1, 20.

Answers to the referees.

None of our referees raised new points after our revision work (see their comments below).

Reviewer #1 (Remarks to the Author):

In this carefully revised manuscript by Besse and coworkers, the authors made a genuine and impressive effort to address the points by all three referees. Overall, although not all experiments worked out for them (not to be expected in the first place), this led to the addition of a series of new display items, both in the main manuscript, in the Supplements and impressively of 4 figures for the referees. Let me select a couple of those new experiments. First, the revised Fig. 5D provides convincing evidence that silencing local circuit activity inhibits local enrichment of arc mRNA suggesting that this might induce/establish subdomains along the axons. This is clearly important and a major addition to the manuscript. Second, I agree with the authors and their new findings in Figure 1 that it does not come as surprise that no major changes in protein levels were found. Of course, I also agree with them that hopefully the SunTag might bring the expected results. A second and important aspect is the extension of the list of synaptic RNAs in their system and the newly added mRNAs, e.g. eIF4A, cg15098 and eIF4E1/bnb and crc. This is clearly additional support for their main findings. Third, I would like to highlight that in the revised version, the fact that Imp-deltaPLD protein is not efficiently transported to axons is important (revised Fig. S3B). I would easily see that this might be even be shown in the main figures. Finally, I like their preliminary findings on arc mRNA presented in Fig. III. This is promising and hope that the observed trend might eventually hold in the future.

Together, this study clearly improved during revision. I am glad that some of the suggested experiments worked out for the authors. Consequently, I suggest publication of this important and timely study, which is a clear advance for the field.

Reviewer #2 (Remarks to the Author):

The authors have been very responsive to suggestions for new experiments and analysis. In particular, new data are provided to validate additional targets identified by transcriptomic analysis. The authors explanations and clarifications are acceptable. This is a very nice paper that will advance the field.

Reviewer #3 (Remarks to the Author):

They now went through an extended revision experimentally addressing the suggestions of the other reviewers and mine. At least in one case (role of compartment-specific dopaminergic innervation), this allowed for the addition of new interesting data. I now support the publication of the manuscript unreservedly.